# BrainDistill: Implantable Motor Decoding with Task-Specific Knowledge Distillation

## Abstract

Transformer-based neural decoders with large parameter counts, pre-trained on large-scale datasets, have recently outperformed classical machine learning models and small neural networks on brain–computer interface (BCI) tasks. However, their large parameter counts and high computational demands hinder deployment in power-constrained implantable systems. To address this challenge, we introduce **BrainDistill**, a novel implantable motor decoding pipeline that integrates a neural decoder with a distillation framework. First, we propose **TSKD**, a task-specific knowledge distillation method that projects task-relevant teacher embeddings into compact student models. Unlike standard feature distillation methods that attempt to preserve teacher representations in full, TSKD explicitly prioritizes features critical for decoding through supervised projection. To evaluate the framework, we define the task-specific ratio (**TSR**), a new metric that quantifies the proportion of task-relevant information retained after projection. Building on this framework, we propose the Implantable Neural Decoder (**IND**), a lightweight transformer architecture that combines linear attention with continuous wavelet tokenization, optimized for on-chip deployment. Across multiple neural datasets, IND consistently outperforms prior neural decoders on motor decoding tasks, while its TSKD-distilled variant further surpasses alternative distillation methods in few-shot calibration settings. Finally, we present a quantization-aware training scheme that enables integer-only inference with activation clipping ranges learned during training. The quantized IND enables deployment under the strict power constraints of implantable BCIs with minimal performance loss.

## 1 Introduction

Decoding movement intentions from neural recordings is a fundamental application of Brain-Computer Interfaces (BCIs), enabling the control of various devices such as drones (Duan et al., 2019), cursors (Bundy et al., 2016), and, most importantly, neural prostheses for patients with disabilities (Lorach et al., 2023). Multiple recording paradigms have been developed to capture neural activity from the motor cortex, including EEG (Cruz-Garza et al., 2014), intracortical spikes (Ma et al., 2017), and Electrocorticography or ECoG (Volkova et al., 2019). Among these, ECoG offers a favorable balance between spatiotemporal resolution and practicality: it provides higher resolution than EEG while remaining more feasible for long-term use compared to invasive intracortical recordings.

A variety of algorithms have been proposed for motor decoding, ranging from classical machine learning methods that rely on hand-crafted features (Wissel et al., 2013; Antuvan & Masia, 2019; Yao et al., 2022) to shallow neural networks that automatically extract task-relevant features from raw signals using convolution and attention mechanisms (Zhang et al., 2022; Song et al., 2022; Mentzelopoulos et al., 2024). More recently, large neural foundation models pre-trained on diverse neural datasets have been introduced to boost decoding performance and improve long-term stability (Zhang et al., 2023; Wang et al., 2023; 2024; Yang et al., 2023). Once pre-trained, these models yield generalizable feature representations that can be efficiently adapted to new datasets with minimal fine-tuning. However, the growing size of neural decoders poses significant challenges for the practical deployment of BCI systems. In fully implantable and long-term medical devices, Systems-on-Chips (SoCs) have been widely adopted (Shao et al., 2025). These SoCs integrate recording, decoding, and potentially stimulation modules into a single cortical implant, making compact and

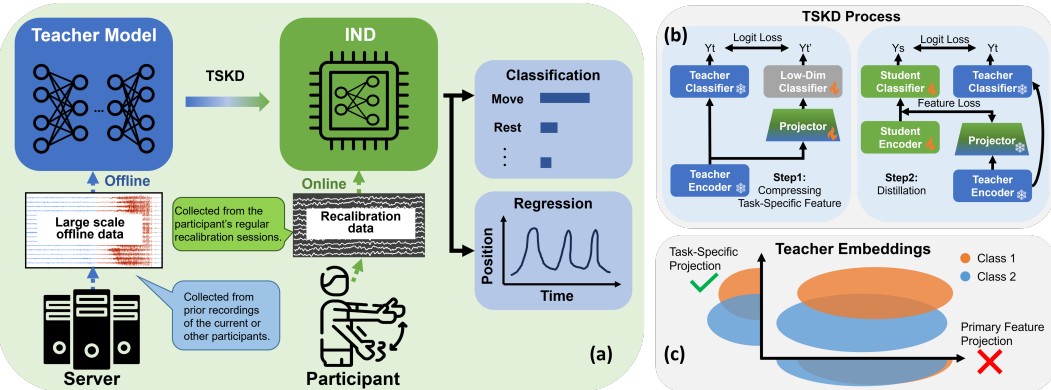

Figure 1: **BrainDistill pipeline:** (a) Training and testing paradigms of BrainDistill. IND distills from a pre-trained large neural model using small-volume recalibration data via TSKD, and then performs online motor decoding. (b) The two-step TSKD process. First, teacher embeddings are compressed in a supervised manner by a projector; second, the fixed projector is used to align student embeddings with the task space. (c) Task-Specific Projection. TSKD projects teacher embeddings into a task-specific subspace, ensuring that critical task-relevant information is well preserved.

energy-efficient decoders essential. For example, Shaeri et al. (2024) developed a brain-to-text decoding chip based on LDA with salient feature selection, while Shin et al. (2022) implemented a tree-based model (Zhu et al., 2020) for brain state classification (e.g., seizures or tremors). The substantial computational complexity of foundation models, combined with the tight hardware constraints of implantable BCI chips, highlights a critical application gap. Although recent work (Lee et al., 2025) significantly reduces the number of trainable parameters of brain foundation models using LoRA (Hu et al., 2022) without sacrificing performance, it still fails to reduce the model size at inference time.

Knowledge distillation (KD) is a popular approach to bridge this gap by transferring knowledge from foundation models to compact student models. However, existing KD methods primarily aim to preserve teacher embeddings as fully as possible (Miles et al., 2024; Zhou et al., 2025; Guo et al., 2023), which becomes problematic when the student model lacks the capacity to mimic complex teacher features, resulting in limited performance gains. In order to address this challenge, we propose **BrainDistill**, a novel implantable motor decoding pipeline that combines a lightweight neural decoder with a task-specific knowledge distillation framework. The decoder achieves strong performance on simple motor decoding tasks and, through distillation from a teacher model, generalizes to more complex tasks. Figure 1 illustrates the BrainDistill pipeline and distillation process. Our main technical contributions are:

- We propose **TSKD**, a novel distillation method for neural decoding that is particularly effective when teacher and student models have a large capacity gap and long-term stable performance must be maintained with minimal recalibration data. Our method prioritizes task-specific information by compressing the teacher embeddings under supervision, thereby compensating for the capacity gap and enabling the student to learn stable features.

- Beyond empirical analysis, we introduce an evaluation protocol to quantify how much the student model can benefit from different projected teacher embeddings. Specifically, we propose a task-specific ratio (**TSR**), a metric that measures how much task-relevant information is preserved after projection, enabling straightforward selection of the best feature projection for any teacher model on a given dataset.

- In addition, we present **IND**, a neural decoder that tokenizes features via the Continuous Wavelet Transform (CWT) and adopts a lightweight, quantization-friendly linear attention architecture. To further reduce hardware cost, we enhance integer-only quantization with a quantization-aware training (QAT) scheme using learnable clipping ranges.

We validate BrainDistill through comparisons with existing decoders, ablations across model architectures and input features, and analysis of chip-level power consumption. The results demonstrate that BrainDistill can serve as a complete implantable motor decoding system with strong poten-

tial for clinical applications. Appendix A.1 reviews prior studies on motor decoding, knowledge distillation, and integer-only quantization, and highlights the novelty of our approach.

## 2 METHOD

### 2.1 PROBLEM DEFINITION

We consider a motor decoding task with neural data $\mathcal{X} = \{(x_1, y_1), (x_2, y_2), \cdots, (x_n, y_n)\}$, where $x_t \in \mathbb{R}^{T \times C}$ represents a segment of brain signals with look back length $T$ and channel number $C$. At time $t$, $y_t$ denotes either the category of movements or the the joint position. A parameterized neural decoder $F$ is composed of a feature extractor $g_\theta$ and a linear classifier layer $f_\phi$. During inference, the model first computes an embedding $z_t = g_\theta(x_t)$, which is then passed to a classifier to produce the prediction $\hat{y}_t = f_\phi(z_t)$. In the simplest setting, $F$ will first train on past recordings $\mathcal{X}_{\text{offline}}$ and then test on new coming neural data $\mathcal{X}_{\text{online}}$. $\mathcal{X}_{\text{online}}$ could contain multiple sessions, where each session is a separate experiment, and the interval between sessions can be days to weeks, depending on the experiment setting. Neural data often drift across recording sessions, which can lead to degraded decoder performance and necessitate periodic recalibration. During recalibration, the parameters $\theta$ and $\phi$ are updated using a subset of recordings $\mathcal{X}_{\text{recalib}}$ from the new session. For efficiency, the size of $\mathcal{X}_{\text{recalib}}$ should be much smaller than that of the offline training set $\mathcal{X}_{\text{offline}}$. Neural foundation models are decoders with millions of parameters that can generate relatively stable embeddings by training on very large $\mathcal{X}_{\text{offline}}$ with self-supervision from multiple sources. During recalibration, only $f\phi$ needs to be updated, and these models outperform conventional decoders when $\mathcal{X}_{\text{offline}}$ is unavailable for the given dataset or when it lacks labels for the target tasks.

BrainDistill is designed to support both paradigms. When a large labeled dataset $\mathcal{X}_{\text{offline}}$ is available, IND can train on it directly; when $\mathcal{X}_{\text{offline}}$ is limited or the task is particularly challenging, IND can instead distill knowledge from neural foundation models using TSKD on the smaller $\mathcal{X}_{\text{recalib}}$.

### 2.2 TASK-SPECIFIC KNOWLEDGE DISTILLATION

When no large $\mathcal{X}_{\text{offline}}$ is available, the decoder often fails to achieve good performance when trained directly on the limited and potentially imbalanced labels of $\mathcal{X}_{\text{recalib}}$. To address this, we distill the logit distribution of $\mathcal{X}_{\text{recalib}}$ from a pretrained teacher model, which enables better generalization to new data. We further introduce a new distillation method that remains effective even under limited-data settings. For the teacher $F_\mathcal{T}$, we denote its embedding on $\mathcal{X}$recalib as $z_\mathcal{T}$, its classifier as $f_\phi^\mathcal{T}$, and its final prediction as $\hat{y}_\mathcal{T}$. We have similar notation for student $F_\mathcal{S}$. The objective of distillation is to minimize $\mathcal{L}_{\text{Distill}} = \|f_\phi^\mathcal{T}(z_\mathcal{T}) - f_\phi^\mathcal{S}(z_\mathcal{S})\|^2$, $z_\mathcal{T} \in \mathbb{R}^{d_t}$, $z_\mathcal{S} \in \mathbb{R}^{d_s}$, $d_s \ll d_t$. We can simplify $\mathcal{L}_{\text{Distill}}$ by focusing only on the weight term of $f_\phi$, which becomes $\mathcal{L}_{\text{Distill}} = \|W_\mathcal{T}^\top z_\mathcal{T} - W_\mathcal{S}^\top z_\mathcal{S}\|^2$, $W_\mathcal{T} \in \mathbb{R}^{d_t \times K}$, $W_\mathcal{S} \in \mathbb{R}^{d_s \times K}$, and $K$ denoting the output dimension of the decoder. For logit-based distillation, both $f_\phi^\mathcal{S}$ and $z_\mathcal{S}$ are randomly initialized and jointly optimized, which often leads to suboptimal convergence. Following prior work, we consider matching not only the logits but also the feature space. Since the dimensions of $d_s$ and $d_t$ do not match, we introduce a projection matrix $P \in \mathbb{R}^{d_t \times d_s}$. The objective is to train $z_\mathcal{S}$ and $P$ such that $z_\mathcal{S} = P^\top z_\mathcal{T}$, thereby transferring task knowledge from teacher embeddings to student embeddings. We rewrite $z_\mathcal{S} = P^\top z_\mathcal{T} + \Delta$, $\Delta$ is the approximation error of the projection, so

$$\mathcal{L}_{\text{Distill}} = \|W_\mathcal{T}^\top z_\mathcal{T} - W_\mathcal{S}^\top (P^\top z_\mathcal{T} + \Delta)\|^2 \tag{1}$$

$$= \|W_\mathcal{T}^\top z_\mathcal{T} - (PW_\mathcal{S})^\top z_\mathcal{T}\|^2 + \|W_\mathcal{S}^\top (P^\top z_\mathcal{T} - z_\mathcal{S})\|^2 + \mathcal{C}(W_\mathcal{S}, z_\mathcal{S}), \tag{2}$$

where the cross-product term is defined as $\mathcal{C}(W_\mathcal{S}, z_\mathcal{S}) = 2\langle (PW_\mathcal{S})^\top z_\mathcal{T} - W_\mathcal{T}^\top z_\mathcal{T}, W_\mathcal{S}^\top (P^\top z_\mathcal{T} - z_\mathcal{S})\rangle$. This shows $\mathcal{L}_{\text{Distill}}$ jointly minimizes two objectives. The first trains a low-rank projection $PW_\mathcal{S}$ to replace $W_\mathcal{T}$, which we refer to as self-compression. The second minimizes the difference between the projected teacher embedding and $z_\mathcal{S}$. The cross-product term encourages both objectives simultaneously. To optimize this process, we first obtain the projection $P^*$ through minimizing self-compression loss $\mathcal{L}_{\text{compress}} = \|W_\mathcal{T}^\top z_\mathcal{T} - (PU)^\top z_\mathcal{T}\|^2$, where $U \in \mathbb{R}^{d_s \times K}$ is the weight matrix of a learnable classifier with the same dimensionality as the student classifier. After that, we freeze $P^*$ and optimize the student model by a joint loss with logit matching and feature matching:

$$\mathcal{L}_{\text{TSKD}} = \mathcal{L}_{\text{Distill}} + \lambda \|P^{*\top} z_\mathcal{T} - z_\mathcal{S}\|^2. \tag{3}$$

Using the same expanding in Eq 1, we can show that

$$\mathcal{L}_{\text{TSKD}} = \|W_{\mathcal{T}}^\top z_{\mathcal{T}} - (P^* W_{\mathcal{S}})^\top z_{\mathcal{T}}\|^2 + \|W_{\mathcal{S}}^\top W_{\mathcal{S}} + \lambda \mathbf{I}\|^2 \|P^{*\top} z_{\mathcal{T}} - z_{\mathcal{S}}\|^2 + \mathcal{C}(W_{\mathcal{S}}, z_{\mathcal{S}}). \quad (4)$$

Here, $P^*$ provides a well-initialized projection, while $\lambda$ serves as a regularizer to stabilize the gradient of the second loss term, enforcing the convergence of $z_{\mathcal{S}}$. Minimizing the objective drives $z_{\mathcal{S}}$ to learn the optimal compressed embedding $P^{*\top} z_{\mathcal{T}}$ for the task.

**Compare with inverse projection.** Existing task-oriented distillation methods also attempt to extract task information from $z_{\mathcal{T}}$ via projection, such as TOFD (Zhang et al., 2020) and TED (Liang et al., 2023). However, an essential difference is that these methods use an inverse projection $P_{\text{inv}} \in \mathbb{R}^{d_s \times d_t}$ and directly minimize the feature loss $\|P_{\text{inv}}^\top z_{\mathcal{S}} - z_{\mathcal{T}}\|^2$. Although the optimal inverse projection $P^*\text{inv}$ under this objective minimizes feature reconstruction error, it may lose task-relevant information when the capacity gap between $z_{\mathcal{T}}$ and $z_{\mathcal{S}}$ is large. In contrast, the optimal $P^*$ obtained through self-compression focuses on task information reconstructing rather than feature reconstruction, as shown in Figure 1(c). In Section 3.3, we further demonstrate the advantage of our projection and show that feature reconstruction is largely uncorrelated with distillation performance.

### 2.2.1 EVALUATINIG TEACHER-STUDENT PROJECTION

As previously mentioned, the distillation loss is determined by the self-compression loss and the embedding loss. It is worth noticing that the embedding loss is constrained by the architecture difference between the teacher and student model, as well as their capacity. Therefore, finding the best $P$ and minimizing the first term will be essential for the distillation performance.

We propose a novel metric to evaluate whether a teacher-student projection matrix $P$ is effective for achieving strong task performance. In order to isolate the effect of $W_{\mathcal{S}}$ and $z_{\mathcal{S}}$, the metric should be a function of given $P$, $W_{\mathcal{T}}$ and teacher embeddings $\mathcal{Z}_{\mathcal{T}} \in \mathbb{R}^{n \times d_t}$ from $\mathcal{X}_{\text{recalib}}$. Starting from the self-compression error $\epsilon_{\text{compress}} = \mathbb{E}_{z \sim \mathcal{Z}_{\mathcal{T}}} \|W_{\mathcal{T}}^\top z - (PU)^\top z\|^2$, we define the metric through its lower bound obtained by optimizing $U$ for the given $P$, as formalized in the following proposition. We prove this proposition in Appendix A.2.

**Proposition 1.** *The compression error $\epsilon_{compress}(U) = \mathbb{E}_{z \sim \mathcal{Z}_{\mathcal{T}}} \|W_{\mathcal{T}}^\top z - (PU)^\top z\|^2$ reaches its minimum $\epsilon_{compress}(U^*) = \left\| \left( I - \Pi_{\mathcal{U}}^{(\Sigma)} \right) W_{\mathcal{T}} \right\|_{\Sigma}^2$ where $U^* = \left( P^\top \Sigma P \right)^\dagger P^\top \Sigma W_{\mathcal{T}}$. $\Pi_{\mathcal{U}}^{(\Sigma)}$ represents the projection to the subspace $\mathcal{U} = span(P)$, where $\Pi_{\mathcal{U}}^{(\Sigma)} = P \left( P^\top \Sigma P \right)^\dagger P^\top \Sigma \in \mathbb{R}^{d_t \times d_t}$. $\Sigma \in \mathbb{R}^{d_t \times d_t}$ is the covariance matrix of $z$.*

Noting that $\left\| \left( I - \Pi_{\mathcal{U}}^{(\Sigma)} \right) W_{\mathcal{T}} \right\|_{\Sigma}^2 + \left\| \Pi_{\mathcal{U}}^{(\Sigma)} W_{\mathcal{T}} \right\|_{\Sigma}^2 = \|W_{\mathcal{T}}\|_{\Sigma}^2$, we observe that $\epsilon_{\text{compress}}$ is controlled by the overlap between the projected student space $\mathcal{U}$ and the teacher classifier. Therefore, we define the task-specific ratio (TSR) of a projection as this overlap proportion:

$$\text{TSR} = \frac{\left\| \Pi_{\mathcal{U}}^{(\Sigma)} W_{\mathcal{T}} \right\|_{\Sigma}^2}{\|W_{\mathcal{T}}\|_{\Sigma}^2} \in [0, 1]. \quad (5)$$

A larger TSR indicates that less task-specific information is lost during projection, yielding a lower $\epsilon_{\text{compress}}$, which in turn serves as the lower bound of the distillation error. We can use TSR to compare different projection methods. We show in the experiments that our supervised projection $P^*$ has the largest TSR compared to PCA or random orthogonal projection.

### 2.3 IND ARCHITECTURE

Our transformer-based decoder adopts a traditional structure, with some specific modification for efficiency. We first calculate spectral-temporal feature of the neural signal through continous wavelet transform (CWT), and tokenize the features in temporal order. These tokens then pass through one linear projection layer and a positional encoding layer. Passing through two linear-attention layers, we obtain the average pooling of the token which serves as the embedding $z$. After the transformer block, we simply use one linear layer for the decoding. In the model, only the last linear layer contains a bias term, while all other linear layers only have weight matrix, which favors the future quantization.

### 2.3.1 TOKENIZATION

For a given sample $x \in \mathbb{R}^{T \times C}$, we first apply z-score normalization over each channel. Then, we compute CWT using complex Morlet wavelet with $N$ different center frequencies, and obtain the spectral-temporal map $x_f \in \mathbb{R}^{N \times T \times C}$. By average pooling through temporal dimension, we obtain a new $x_f \in \mathbb{R}^{N \times L \times C}$, where $L \ll T$. Then the feature is tokenized into $x_{\text{input}} \in \mathbb{R}^{L \times (C*N)}$, where each token contains both channel and frequency information from different time step. The token is projected to embedding space through $x_{\text{emb}} = x_{\text{input}} W + \text{Pos}$, where $x_{\text{emb}} \in \mathbb{R}^{L \times d}$, $W \in \mathbb{R}^{(C*N) \times d}$ is a linear transform, and $\text{Pos} \in \mathbb{R}^{L \times d}$ is positional embedding. Using CWT features as tokens offers clear advantages: each token encodes explicit time–frequency information within a specific segment and frequency, making the features directly interpretable. This is particularly valuable for ECoG, which can precisely reveal the frequency bands of neural activity. In contrast, convolutional tokenization provides only implicit interpretability, as its time–frequency representation depends on kernel size and learned weights. We present a comparison of CWT- and convolution-based features in Appendix A.7, and further analyze the contribution of each CWT frequency to decoding performance.

### 2.3.2 LINEAR ATTENTION

We customize the linear attention method from Katharopoulos et al. (2020) for neural data and the need for quantization. In original linear attention, $K$ and $V$ are first multiplied because $L$ can be much larger than $d$, but for neural data, $L$ is short as only recent neural data is meaningful for predicting the movement. Therefore, we compute linear attention with the same order as softmax attention: $V_i' = \frac{\left[ \sum_{j=1}^N \phi(Q_i)^\top \phi(K_j) \right] V_j}{\sum_{j=1}^N \phi(Q_i)^\top \phi(K_j)}$. $V_i'$ is the $i$-th token of attention output, and $\phi(\cdot)$ is the activation function, where we use ReLU for its quantization-friendly properties.

## 2.4 QUANTIZATION-AWARE TRAINING WITH INTEGER-ONLY INFERENCE

We develop an integer-only quantization-aware training (QAT) pipeline with learnable clipping ranges to ensure that IND can be deployed on a chip with minimal performance loss. Unlike previous integer-only quantization approaches which estimate clipping ranges statistically, such as I-ViT (Li & Gu, 2023), our QAT scheme introduces learnable clipping ranges that are jointly optimized with the network.

First, the input neural signal is quantized through a lookup table that maps $x_t$ to unsigned 10-bit data. This simulates the function of a 10-bit analog digital converter (ADC). The Wavelet kernels use 16-bit symmetric uniform quantization; wavelet features are 8-bit. In the decoder, all weights and activations are 8-bit, and biases are 32-bit. During QAT, we update model weights as well as the clipping range $\alpha$ for activation values. We define a parameterized $\alpha_l$ for a certain layer $l$ of the model, thus the scaling factor for quantization is $s_{l+1} = \frac{\alpha_l}{2^{b-1}-1}$ if the quantization bit-width is $b$. Given the activation $y_l$ and its scale $s_l$, the quantized activation will be computed through:

$$y_l^q = \left\lfloor y_l * \frac{s_l}{s_{l+1}} \right\rceil * s_{l+1}, \tag{6}$$

where $s_{l+1}$ is computed by $\alpha_l$. Inspired by PACT (Choi et al., 2018), we optimize $\alpha_l$ through assigning gradients according to $y_l^q$ when it falls out of the clipping range $[-\alpha_l, \alpha_l]$, which is:

$$\frac{\partial y_l^q}{\partial \alpha_l} = \frac{\partial y_l^q}{\partial y_l} \frac{\partial y_l}{\partial \alpha_l} = \begin{cases} -1, & x \in (-\infty, \alpha_l) \\ 1, & x \in [\alpha_l, +\infty) \end{cases}. \tag{7}$$

$x$ is the element of $y_l^q$, and we set $\frac{\partial y_l^q}{\partial y_l} = 1$ by Straight-Through Estimator (STE) Bengio et al. (2013). We implement LayerNorm using the same approach as I-ViT (Li & Gu, 2023), and perform an approximation for the division operation in linear attention. The implementation details of quantization are in Appendix A.6.1. In order to avoid any floating-point computation during inference time, $\frac{s_l}{s_{l+1}}$ will be converted to a dyadic number $\frac{s_l}{s_{l+1}} \approx \frac{m}{2^e}$, where $m$ and $e$ are 16-bit integers. Although $\alpha$ can be updated during training, its initialization remains important. We estimate it from the activation distribution of the training set on the full-precision model. During QAT, the quantized

Table 1: Summary of datasets. #S represents the number of subjects, and #Ses is the average number of sessions per subject. In test cases, **Scratch** refers to training the decoder directly on $\mathcal{X}_{\text{offline}}$ or $\mathcal{X}_{\text{recalib}}$, while **Distillation** refers to training the decoder via distillation from a pre-trained teacher model on $\mathcal{X}_{\text{recalib}}$.

| Dataset | Type | Source | #S | #Ses | Paradigm | Teacher |
|---------|------|--------|-----|------|----------|---------|
| BCIC-2A | EEG | Tangermann et al. (2012) | 9 | 1 | Distillation | EEGPT (Wang et al., 2024) |
| BCIC-2B | EEG | Steyrl et al. (2016) | 9 | 1 | Distillation | EEGPT (Wang et al., 2024) |
| FALCON-M1 | Spikes | Karpowicz et al. (2024) | 1 | 5 | Distillation | NDT2 (Ye et al., 2023) |
| Human-D | ECoG | Peterson et al. (2021) | 12 | 7 | Scratch | / |
| Monkey-R | ECoG | Private | 1 | 12 | Scratch | / |
| Human-C | ECoG | Private | 1 | 6 | Scratch Distillation | Transformer |

decoder is initialized from the full-precision model and then further optimized using either the task loss or the distillation loss.

## 3 EXPERIMENTS

We evaluate BrainDistill on multiple human and non-human primate ECoG datasets spanning a comprehensive suite of upper-limb decoding tasks, including naturalistic movement detection, fine-grained movement classification, and trajectory decoding. We consider two training paradigms—directly training on $\mathcal{X}_{\text{recalib}}$ (**Scratch**) and distillation from pretrained models (**Distillation**) —and compare projection methods for knowledge distillation, demonstrating the advantage of our approach. To assess cross-modal generalizability, we further test TSKD on EEG and intracortical spike recordings. In addition, we analyze the performance and energy consumption of the quantized IND, highlighting its feasibility for on-chip implementation in BCI implants. Detailed settings are provided in A.3.

### 3.1 EXPERIMENTAL SETTINGS

Table 1 summarizes the datasets and tasks, including two private ECoG datasets, **Human-C** and **Monkey-R**. Further details are provided in Appendix A.5. For baseline methods, we first compare our decoder trained from scratch with other established neural decoding models, including EEGNet (Lawhern et al., 2018), EEGConformer (Song et al., 2022), ATCNet (Altaheri et al., 2023) and CT-Net (Zhao et al., 2024).We also compare with neural foundation model LaBraM (Jiang et al., 2024) with linear probing. Secondly, we compare our distillation method with other established logit-based, feature-based and task-oriented knowledge distillation methods: KD (Hinton et al., 2015), SimKD (Chen et al., 2022), VkD (Miles et al., 2024), RdimKD (Guo et al., 2023), TOFD (Zhang et al., 2020) and TED (Liang et al., 2023). During distillation, we leverage three different pre-trained foundation models based on the task, EEGPT (101M parameters) (Wang et al., 2024), NDT2 (3.7M parameters) (Ye et al., 2023) and a 100M transformer pre-trained specifically on **Human-C**.

### 3.2 DISTILLED MULTI-CLASS MOTOR DECODING

We test both **Scratch** and **Distillation** on **Human-C** under intra-session and inter-session scenarios. Relative to the starting date of the first session, the 6 sessions are 1, 4, 5, 6, 16 and 17, which are divided into 5 train-test splits: $[1-1, 4-4, 4-5, 4-6, 16-17]$. Session 1 and 4 are divided into training and test parts, and 5, 6 and 17 only contain test parts, 16 only contains training parts. Therefore, notation $1-1$ means training on the training split of session 1, and testing on the testing split of session 1. The teacher model consists of a large transformer encoder (100M parameters) with self-supervised pre-training through a masked reconstruction approach. The teacher encoder is trained on 159 previous sessions, and then it is fixed, and only the classifier $f_\phi$ is tuned across the sessions. The split is based on which samples are used to update $f_\phi$. Since the participant performs different movements in each session, the split $1-1$ and $16-17$ contains 4 categories, and $4-4, 4-5, 4-6$ contains 6 categories. In experiments we test only using the $\mathcal{L}_{\text{TSKD}}$ (+TSKD) and also distillation loss combined with task loss, which is cross entropy loss here (+TSKD(CE)).

Table 2: Movement decoding performance on **Human-C**. The number of parameters are indicated for each model. The For each split, we report the mean and standard deviation of F1 and Avg Recall over three random initializations. Models are selected based on a validation set comprising 20% of the training data. Scores are reported in %. Our method and the best results are shown in **bold**, with the second-best results underlined.

| Session | 1-1 | | 4-4 | | 4-5 | | 4-6 | | 16-17 | |
|---|---|---|---|---|---|---|---|---|---|---|
| Metrics | F1 | Avg Recall | F1 | Avg Recall | F1 | Avg Recall | F1 | Avg Recall | F1 | Avg Recall |
| Teacher (100M) | 77.6 | 67.4 | 73.3 | 57.2 | 74.5 | 60.8 | 77.3 | 62.5 | 80.6 | 58.9 |
| Conformer (630K) | 37.7 ±11.1 | 38.3 ±4.36 | 40.8 ±5.21 | 31.5 ±2.79 | 36.0 ±8.45 | 33.9 ±13.9 | 39.9 ±6.39 | 23.7 ±1.61 | 43.1 ±7.58 | 41.4 ±5.05 |
| ATCNet (114K) | 44.9 ±3.80 | 33.9 ±3.90 | 48.4 ±5.39 | 26.9 ±3.37 | 45.2 ±1.94 | 22.4 ±2.59 | 54.4 ±3.95 | 23.2 ±2.03 | 43.7 ±3.10 | **53.8 ±1.29** |
| CTNet (152K) | 46.1 ±0.40 | 30.5 ±1.11 | 49.1 ±2.25 | 27.2 ±1.57 | 42.2 ±2.25 | 23.5 ±1.45 | 54.3 ±0.77 | 21.6 ±2.04 | 44.3 ±3.20 | 49.7 ±3.29 |
| EEGNet (5.5K) | 29.0 ±16.8 | 39.2 ±4.57 | 42.9 ±1.89 | 28.8 ±3.75 | 41.8 ±4.76 | 25.1 ±7.98 | 42.8 ±6.38 | 17.5 ±1.59 | 54.1 ±9.01 | 43.7 ±6.43 |
| LaBraM (5.8M) | 44.9 ±5.35 | 28.1 ±1.85 | 57.8 ±0.95 | 34.3 ±2.14 | 56.9 ±1.02 | 34.4 ±1.24 | 60.9 ±0.77 | 25.9 ±1.18 | 71.9 ±2.21 | 37.9 ±2.89 |
| **IND (30K)** | **69.1 ±1.64** | **56.6 ±4.11** | **64.9 ±3.45** | **44.9 ±3.78** | **61.5 ±2.20** | **37.1 ±5.68** | **70.6 ±0.97** | **44.2 ±5.26** | **72.3 ±0.83** | 52.1 ±5.77 |
| IND + KD | 68.5 ±2.56 | 58.5 ±1.90 | 67.8 ±0.49 | 50.2 ±1.48 | 61.1 ±1.72 | 41.2 ±0.88 | 71.9 ±1.19 | 48.0 ±3.01 | 66.6 ±2.48 | 61.9 ±2.62 |
| IND + SimKD | 61.7 ±1.70 | 44.7 ±1.64 | 69.1 ±0.33 | 50.1 ±0.18 | 65.8 ±1.67 | 45.8 ±2.55 | 72.8 ±1.00 | 46.3 ±2.67 | 68.4 ±0.32 | 43.7 ±0.86 |
| IND + VkD | 70.9 ±1.32 | 60.7 ±1.49 | 67.6 ±0.80 | 50.1 ±0.54 | 62.2 ±2.68 | 41.5 ±3.00 | 71.7 ±1.66 | 49.7 ±2.10 | 69.9 ±2.35 | 63.4 ±2.49 |
| IND + RdimKD | 72.3 ±1.84 | 63.1 ±2.58 | 68.6 ±0.76 | 52.1 ±0.99 | 63.1 ±2.49 | 41.9 ±3.02 | 72.3 ±1.46 | 47.9 ±0.82 | 65.9 ±0.38 | 63.4 ±2.96 |
| IND + TOFD | 71.2 ±0.75 | 62.7 ±1.86 | 68.1 ±0.91 | 50.2 ±1.44 | 61.9 ±1.54 | 41.5 ±3.64 | 72.6 ±0.24 | 46.6 ±2.89 | 67.6 ±3.48 | **66.9 ±1.72** |
| IND + TED | 71.2 ±0.98 | 63.6 ±0.42 | 67.9 ±1.13 | 50.8 ±1.45 | 61.5 ±0.59 | 42.2 ±2.03 | 71.9 ±1.01 | 48.9 ±3.18 | 69.1 ±3.02 | 64.9 ±1.00 |
| **IND + TSKD(CE)** | 73.4 ±0.75 | **65.2 ±5.67** | 68.1 ±0.77 | 51.7 ±0.80 | 65.1 ±1.00 | 47.7 ±3.71 | 71.7 ±3.28 | 52.4 ±1.59 | 69.8 ±1.47 | **65.5 ±1.94** |
| **IND + TSKD** | **73.9 ±1.64** | 64.9 ±0.39 | **73.3 ±0.63** | **58.0 ±0.98** | **75.0 ±0.20** | **60.9 ±0.61** | **77.3 ±0.34** | **59.6 ±0.70** | **70.4 ±1.28** | 56.3 ±1.34 |

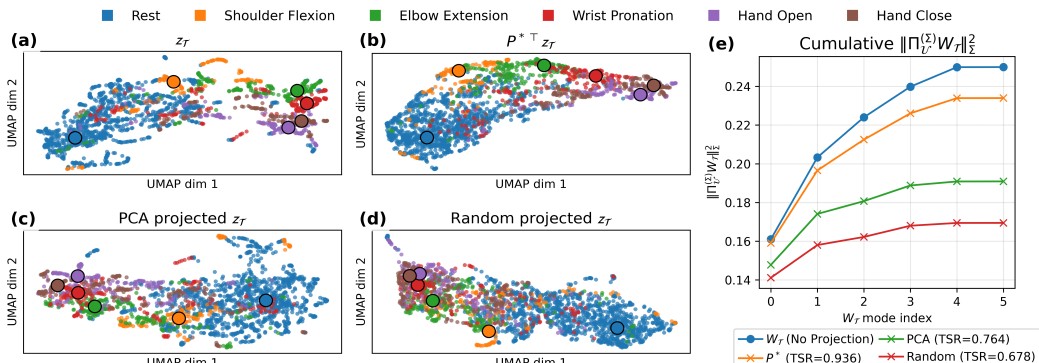

Figure 2: **Visualization of different projections on Human-C:** (a) UMAP visualization of teacher embeddings. (b) UMAP visualization of teacher embeddings projected by $P^*$. (c) UMAP visualization of teacher embeddings projected by PCA. (d) UMAP visualization of teacher embeddings projected by a random orthogonal matrix. Each color denotes a movement, and the density centers of each class are highlighted. (e) The cumulative value $\left\|\Pi_{\mathcal{U}}^{(\Sigma)} W_{\mathcal{T}}\right\|_{\Sigma}^2$ for each projection, computed over 6 task-space dimensions and accumulated. Higher values indicate that the projected features are more likely to yield better decoding performance. For reference, the cumulative $\|W_{\mathcal{T}}\|_{\Sigma}^2$ of the original feature space is shown in blue, corresponding to TSR = 1.

Table 2 shows the classification performance across different sessions of **Human-C**. We report both F1 score and the average Recall value. Given that the trials of different movements are highly imbalanced, as resting states are always the majority class, average Recall reveals better model capability to decode different movements. According to the results, our decoder significantly outperforms other decoders when trained from scratch, but still has a large gap compared to the teacher model. Besides, LaBraM outperforms other baselines but is still significantly wrose than IND, which further verifies the design choice. With knowledge distillation, the performance keeps improving, and TSKD has the best performance in $4-4$, $4-5$ and $4-6$, as the teacher itself offers great enough predictions in these sessions. In $1-1$ and $16-17$ TSKD(CE) has the best performance, mainly because of the lower quality of teacher predictions, especially in $16-17$. We further show that TSKD generalizes to other decoder architectures by applying it to EEGConformer, ATCNet and CTNet, the results are presented in Appendix A.5.1.

### 3.3 Measuring Teacher-Student Projection

As discussed in Section 2.2.1, we propose TSR to evaluate how well a projection $P$ preserves task information in teacher embeddings, which directly impacts student model performance. We first

Table 3: TSKD performances with different projections and inverse projection on **Human-C**. For each split, we report the mean and standard deviation of F1 and Avg Recall over three random initializations. The models for testing are selected based on a validation set of 20% training data. Scores are in %. Our method name and the best performances are **bolded**, and the second best are underlined.

| SESSION | 4-4 | | 4-5 | | 4-6 | |
|---|---|---|---|---|---|---|
| METRICS | F1 | Avg Recall | F1 | Avg Recall | F1 | Avg Recall |
| Teacher Model | 73.3 | 57.2 | 74.5 | 60.8 | 77.3 | 62.5 |
| Inverse Projection | 73.1 ±0.41 | 57.6 ±1.24 | 71.7 ±1.27 | 55.2 ±0.92 | 75.5 ±0.94 | 56.7 ±1.33 |
| **TSKD** | **73.3 ±0.63** | **58.0 ±0.98** | **75.0 ±0.20** | **60.9 ±0.61** | **77.3 ±0.34** | **59.6 ±0.70** |
| TSKD(PCA) | 71.8 ±0.92 | 55.4 ±2.02 | 72.5 ±0.41 | 56.0 ±0.41 | 77.3 ±1.02 | 57.6 ±2.51 |
| TSKD(Random) | 71.7 ±0.37 | 55.1 ±0.50 | 73.8 ±1.57 | 57.9 ±2.32 | 76.9 ±0.45 | 57.7 ±2.48 |

Table 4: Correlation between projection evaluation metrics and task performances on three different projections (TSKD, PCA and Random) on **Human-C**.

| SESSION | 4-4 | | 4-5 | | 4-6 | |
|---|---|---|---|---|---|---|
| METRICS | F1 | AVG RECALL | F1 | AVG RECALL | F1 | AVG RECALL |
| **TSR** | **0.9908** | **0.9891** | **0.9999** | **0.9783** | **0.9167** | **0.9241** |
| Mutual Information | -0.4408 | -0.1721 | -0.4336 | -0.2310 | -0.7854 | 0.0061 |
| Relative Reconstruction Error | 0.3524 | 0.0765 | 0.3431 | 0.1345 | 0.7750 | -0.0227 |

compare our projection $P^*$ with PCA projection and random orthogonal projection based on distillation performance. By replacing $P^*$ with PCA or random projection, we test the model on $4-4$, $4-5$ and $4-6$ of **Human-C**. To demonstrate the necessity of compressing task-specific teacher embeddings for small student models, we also compare our projection with an inverse projection baseline. In the inverse projection setting, IND passes its embeddings through an inverse projector $P_{\text{inv}} \in \mathbb{R}^{d_t \times d_s}$ and obtains projected student embeddings $z_{\mathcal{S}}^p = P_{\text{inv}}^\top z_{\mathcal{S}} \in \mathbb{R}^{d_t}$, then $z_{\mathcal{S}}^p$ is aligned with $z_{\mathcal{T}}$. These inversely projected embeddings are directly used for classification. However, due to the large dimensionality gap between teacher and student embeddings, the inverse projector significantly increases the parameter count of IND. As shown in Table 3, our projection achieves better distillation performance. Interestingly, TSKD even outperforms inverse projection, although the latter aligns embeddings in a higher-dimensional space. This result suggests that inverse projection cannot overcome the inherent feature bottleneck of the student model, and therefore direct alignment in the teacher space still leads to a loss of task-relevant information.

We visualize the learned embeddings from session 5 under different projections, and report their cumulative $\|W_{\mathcal{T}}\|_\Sigma^2$ and TSR values. In Figure 2, we visualize teacher embeddings and different projected embeddings through UMAP McInnes et al. (2018). According to the results, $P^*$ produces more separable embeddings than other projections, even compared to $z_{\mathcal{T}}$. This can be explained by the fact that when the high-dimensional teacher features are down-projected in an unsupervised manner, much of the task-related information is lost, as illustrated in Figure 2 (a), (c), and (d). In contrast, $P^*$ prioritizes task-relevant information over other variations, resulting in a more separable subspace, which is consistent with both Figure 2 (b) and the TSR values.

To further quantify that TSR is capable of diagnosing projections for knowledge distillation, we compute the correlation between TSR metrics and the task performances of different projections on three sessions of **Human-C**. A high absolute correlation indicates that TSR is a task-specific metric for projection quality and can be used to predict downstream task performance. We also compute the correlations for conventional metrics, including mutual information and relative reconstruction error between the projected embeddings and the teacher embeddings. The details of how these metrics are computed, along with their values for different projections, are provided in Appendix A.4. Results in Table 4 show that TSR achieves a correlation above 0.9 with task performance across sessions, whereas mutual information and relative reconstruction error exhibit almost no correlation. These results further validate the design of both TSR and TSKD.

### 3.4 MOTOR DECODING WITHOUT TEACHER MODEL

We further test how IND performs and generalizes across sessions and subjects on simpler tasks when no teacher models are available. We perform joint trajectory regression on **Monkey-R** and binary movement state detection on **Human-D**. On **Monkey-R**, we train our model on the first 5

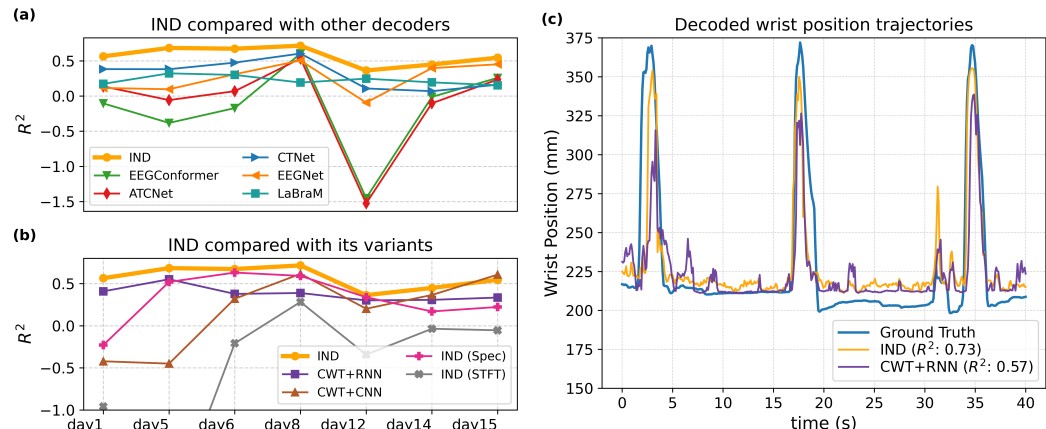

Figure 3: **Results of movement regression on Monkey-R.** (a) $R^2$ values of IND and different baseline decoders across seven test sessions. (b) $R^2$ values of IND and its variants with different tokenization methods or model architectures; exact $R^2$ values are reported in Appendix A.5.2. (c) Illustration of decoding output: comparison of IND and CWT+RNN on a 40-second segment from day 6, with corresponding $R^2$ values.

Table 5: Movement detection performance on **Human-D**. The average accuracy and AUROC over 12 patients are reported. Scores are in %. Our method name and the best performances are **bolded**, and the second best are underlined.

| SETTING | METRIC | **IND** | CONFORMER | ATCNET | CTNET | EEGNET | LABRAM |
|---|---|---|---|---|---|---|---|
| Intra-Subject | Acc | 77.1 | 71.6 | 62.1 | 67.6 | 55.1 | **77.4** |
| | AUROC | **85.0** | 80.9 | 67.2 | 72.0 | 61.6 | **85.0** |
| Inter-Subject | Acc | 54.5 | 51.8 | 51.3 | 51.1 | 50.1 | **55.0** |
| | AUROC | **59.8** | 54.9 | 52.9 | 55.4 | 54.2 | 56.9 |

sessions, picking the best weights based on a 20% split of the validation set, and test on the remaining 7 sessions. Besides baseline decoders, we also perform an ablation study on the backbone model and input features of IND, replacing the transformer with CNN (CWT+CNN) or RNN (CWT+RNN), and replacing CWT tokens with spectrogram (IND (Spec)) or Short-Time Fourier Transform (IND (STFT)). We can see from Figure 3 (a) that our decoder has the best performance across 15 days. Figure 3 (b) shows that our decoder precisely captures the onsets and offsets of wrist movements. We can see from Figure 3 (c) that IND better captures the movement onsets and offsets, while CWT+RNN does less well and has more false spikes in the prediction. We also conducted an online decoding experiment on day 5, which is described in detail in Appendix A.5.2.

On **Human-D**, the task is to detect whether the arms of subjects are moving or resting. We evaluate both intra-subject and inter-subject performance. For intra-subject, we train on each subject individually, and divide the last session as the test set for each one. For inter-subject, we still test on the last session of each subject, but we train on all other subjects. Table 5 shows that our decoder has a similar performance as LaBraM with linear probing, and outperforms other models. In general, the results on **Monkey-R** and **Human-D** support that our decoder can achieve great performance on simple decoding tasks without distillation and maintain a recognizable stability over time. We perform an ablation study of IND on **Human-D** as well, and present results in Appendix A.5.3.

### 3.5 FEASIBILITY OF ON-CHIP IMPLEMENTATION

We quantize IND with an architecture tailored for **Human-C** and **Monkey-R**. The decoder processes 320-dimensional input features, corresponding to either 5 wavelet features over 64 channels or 10 wavelet features over 32 channels, and includes 2 attention layers with 32-dimensional embeddings. We compare full-precision and quantized performance on **Human-C**, and estimate power consumption based on the number of operations, model weights, and I/O memory (see Appendix A.6.2). In order to ensure the safety of the participant, the implanted BCI has to dissipate less than 15-40 mW of power, to avoid heating surrounding brain tissue by more than $2°C$, which causes cellular damage (Kim et al., 2007; Wolf, 2011; Karageorgos et al., 2020). Table 6 shows that our quantized decoder significantly outperforms I-ViT over 3 sessions, and loses < 3% of performance compared to FP32,

Table 6: Comparison of decoding performance on **Human-C** between the full precision model (FP32), I-ViT (W8A8) and our quantization method (W8A8). For each split, we report F1 and Avg Recall. The estimated power consumption (mW) is reported in the last column. Our results are **bolded**.

| METHOD | 1-1 | | 4-4 | | 4-5 | | 4-6 | | 16-17 | | POWER (mW) |
|---|---|---|---|---|---|---|---|---|---|---|---|
| METRICS | F1 | Avg Recall | F1 | Avg Recall | F1 | Avg Recall | F1 | Avg Recall | F1 | Avg Recall | |
| FP32 | 73.9 | 64.9 | 73.3 | 58.0 | 75.0 | 60.9 | 77.3 | 59.6 | 70.4 | 56.3 | 22.84 |
| I-ViT | 72.4 | 62.0 | 73.4 | 58.3 | 71.0 | 53.9 | 74.5 | 55.7 | 73.5 | 46.7 | 5.66 |
| **Ours** | **72.1** | **62.1** | **73.5** | **57.7** | **73.2** | **57.6** | **77.8** | **58.7** | **72.6** | **55.2** | **5.66** |

Table 7: Distilled performance on two EEG datasets (top) and a spike dataset from FALCON benchmark (bottom). For **BCIC-2A**, the average weighted F1 over 9 subjects is reported; for **BCIC-2B**, the average AUROC over 9 subjects is reported. For **FALCON-M1**, the average $R^2$ of 16-channel EMG signals is reported. We report the mean and std of scores over 3 random initializations. Scores are in %. Our method name and the best performances are **bolded**, and the second best are underlined. / means the loss is not applicable to the dataset.

| DATASET | TEACHER | IND | +KD | +SIMKD | +VKD | +RDIMKD | +TOFD | +TED | **+TSKD** | **+TSKD(CE)** |
|---|---|---|---|---|---|---|---|---|---|---|
| BCIC2A | 50.66 | 24.22 | 24.95 ±0.77 | 15.98 ±1.12 | 25.86 ±0.88 | 23.91 ±1.37 | 23.88 ±3.49 | 23.67 ±3.12 | 26.12 ±1.56 | **26.73 ±2.44** |
| BCIC2B | 80.51 | 68.79 | 67.76 ±0.56 | 57.91 ±1.51 | 67.29 ±0.81 | 69.78 ±0.38 | 67.59 ±0.56 | 67.42 ±0.60 | **70.33 ±0.64** | 69.78 ±0.48 |
| FALCON-M1 | 81.95 | 37.59 | 41.46 ±1.15 | 37.30 ±3.07 | 40.73 ±3.76 | 41.57 ±0.69 | 39.21 ±2.15 | 34.83 ±7.36 | **43.52 ±1.53** | / |

yet saves more than $3\times$ of power consumption. The result indicates that it is applicable to implant IND on a cortical BCI device.

## 3.6 DISTILLED MOTOR DECODING ACROSS RECORDING MODALITIES

Besides ECoG, we also evaluate TSKD with our decoder on other common data modalities for movement decoding, such as EEG and intracortical spikes. We test on EEG datasets **BCIC-2A** and **BCIC-2B** (Tangermann et al., 2012; Steyrl et al., 2016), where **BCIC-2A** is a 4-class classification task and **BCIC-2B** is binary. We use EEGPT (Wang et al., 2024) as the teacher model, fine-tune its classifier on the full training set, and extract an imbalanced subset from it as $\mathcal{X}_{\text{recalib}}$. We also test **FALCON-M1** dataset from FALCON benchmark (Karpowicz et al., 2024), using NDT2 (Ye et al., 2023) as the teacher model. Similarly, the teacher model is fine-tuned on the whole training set, while only a fraction of it is leveraged as $\mathcal{X}_{\text{recalib}}$. According to Table 7, only TSKD and TSKD(CE) consistently improve decoder performance, while other distillation methods sometimes reduce the decoder performance. The results confirm that TSKD is also effective in other data modalities. We also compare different distillation approaches with EEGConformer on BCIC2A and BCIC2B to verify the cross-model and cross-modality generalization ability of TSKD, and we present the results in Appendix A.5.4.

## 4 CONCLUSION

In this paper, we introduce BrainDistill, a motor decoding pipeline that integrates a task-specific knowledge distillation (TSKD) method with a transformer-based implantable neural decoder (IND). TSKD projects task-relevant teacher embeddings into compact student models, explicitly prioritizing features that are most critical for decoding performance, and its effectiveness is quantified by the task-specific ratio (TSR) metric. IND is a lightweight transformer decoder tailored for on-chip deployment, which combines linear attention with continuous wavelet transform tokenization to extract interpretable time–frequency features. To ensure deployability under strict hardware constraints, we also introduce a quantization-aware training strategy that learns activation clipping ranges and enables efficient integer-only inference. Extensive experiments across human and non-human primate ECoG, as well as other recording modalities, demonstrate that BrainDistill not only outperforms prior neural decoders but also surpasses alternative distillation baselines in few-shot calibration settings. It also achieves better generalization across sessions and subjects. With its compact design, strong decoding performance, and chip-ready implementation, BrainDistill takes a practical step toward clinical deployment of neural network–based BCI systems.

## 5 ETHICAL STATEMENTS

The collection of and experiments on **Human-C** and **Monkey-R** in our study were approved by the Institutional Review Board (IRB) and underwent ethical review. Informed consent was obtained from all participants prior to experimental procedures. The data were collected as part of a clinical trial aimed at developing motor decoding algorithms to restore upper-limb motor function after spinal cord injury. The data collection is conducted in accordance with the Declaration of Helsinki.

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

# A APPENDIX

## A.1 RELATED WORKS

**Neural Decoders for ECoG Movement Decoding.** Prior to our work, Hammer et al. (2013) showed that time–frequency features of ECoG signals can decode continuous movement trajectories, speed, and acceleration, and proposed a multiple linear regression decoder using short-term Fourier transform (STFT) features. Flint et al. (2020) demonstrated that ECoG enables precise decoding of finger movements and forces by leveraging LFADS Pandarinath et al. (2018), a sequential deep learning model. Eliseyev et al. (2017) introduced REW-NPLS, a multilinear regression method for decoding upper-limb movements from continuous wavelet transform (CWT) features, while Śliwowski et al. (2022) replaced the multilinear model with CNN and LSTM architectures, achieving improved results. These studies establish the effectiveness of time–frequency features, especially CWT features, for ECoG-based movement decoding; however, transformer-based decoders that directly tokenize CWT features have not yet been explored.

**Knowledge Distillation.** To improve distillation from teacher models, Chen et al. (2022) projected student features into the teacher space, allowing the student to reuse the teacher's classifier and achieve comparable performance. Miles & Mikolajczyk (2024) further analyzed the role of projectors in distillation and found that simpler projections improve the correlation between student and projected features, leading to better results with orthogonal projection (Miles et al., 2024). Beyond learnable projections, Zhou et al. (2025); Guo et al. (2023) directly applied PCA to teacher features and aligned student features with the projected ones. However, these approaches preserve teacher information indiscriminately, which may not be optimal for specific tasks. For task-oriented distillation, Zhang et al. (2020) proposed logit distillation with additional FC layers to implicitly align features, and Liang et al. (2023) aligned teacher and student intermediate features through task filters at each layer. Such methods, however, require architectural similarity between teacher and student models and cannot ensure the effectiveness of the implicit alignment. In contrast, our TSKD explicitly compresses task-relevant information from teacher embeddings and quantifies the projection quality using a novel metric. Besides distillation-based methods, recent work (Lee et al., 2025) shows that pre-trained neural foundation models can achieve downstream performance comparable to full fine-tuning while requiring only a small number of trainable parameters through LoRA (Hu et al., 2022). However, these PEFT-based approaches do not reduce the model size at inference time and are therefore unsuitable for implantable decoding scenarios.

**Integer-Only Quantization.** To enable hardware implementation of neural networks, integer-only quantization methods have been proposed. Unlike conventional quantization, these methods ensure that all operations are executed entirely in the integer domain, which requires both non-linear functions and scaling factors to be quantized. For example, I-BERT (Kim et al., 2021) and FQ-ViT (Lin et al., 2021) replace non-linear operations in transformers, while I-ViT (Li & Gu, 2023) further fully quantizes the entire computational graph of ViTs using integer arithmetic and bit-shifting. Building on this line of work, I-LLM (Hu et al., 2024) introduces DI-MatMul, which dynamically adjusts the clipping range for each token. However, none of these approaches integrate learnable quantization clipping ranges with integer-only inference in a quantization-aware training framework. This integration is particularly crucial for small models, where QAT is essential to jointly optimize model weights and clipping ranges to mitigate quantization errors. To the best of our knowledge, our approach is the first to achieve this.

## A.2 PROOFS

PROOF OF PROPOSITION 1

**Setup.** Let $W_{\mathcal{T}} =: W \in \mathbb{R}^{d_t \times K}$, $P \in \mathbb{R}^{d_t \times d_s}$, $U \in \mathbb{R}^{d_s \times K}$. Assume $z$ has zero mean (otherwise classifier biases absorb the mean), and let $\Sigma = \mathbb{E}[zz^\top] \succeq 0$. Define the $\Sigma$-weighted Frobenius norm

$$\|A\|_\Sigma^2 = \mathrm{tr}(A^\top \Sigma A).$$

Then the compression error is

$$\epsilon_{\mathrm{compress}}(U) = \mathbb{E}\left\|W^\top z - (PU)^\top z\right\|^2 = \mathrm{tr}\left((W - PU)^\top \Sigma (W - PU)\right) =: L(U).$$

**Step 1.** Expanding $L(U)$ and differentiating with respect to $U$ yields the normal equations

$$\nabla_U L(U) = 2\big(P^\top \Sigma P U - P^\top \Sigma W\big) = 0 \quad \Longleftrightarrow \quad \big(P^\top \Sigma P\big) U = P^\top \Sigma W.$$

If $P^\top \Sigma P$ is invertible, the unique minimizer is

$$U^* = \big(P^\top \Sigma P\big)^\dagger P^\top \Sigma W,$$

$A^\dagger$ is the pseudoinverse of $A$ in rank-deficient case.

**Step 2.** Define the $\Sigma$-orthogonal projector onto $\mathcal{U} = \mathrm{span}(P)$ by

$$\Pi_{\mathcal{U}}^{(\Sigma)} = P\big(P^\top \Sigma P\big)^\dagger P^\top \Sigma \ \in \ \mathbb{R}^{d_t \times d_t}.$$

Using the expression above for $U^*$,

$$PU^* = P\big(P^\top \Sigma P\big)^\dagger P^\top \Sigma W = \Pi_{\mathcal{U}}^{(\Sigma)} W.$$

It holds that $\big(\Pi_{\mathcal{U}}^{(\Sigma)}\big)^2 = \Pi_{\mathcal{U}}^{(\Sigma)}$ and $\big(\Pi_{\mathcal{U}}^{(\Sigma)}\big)^\top \Sigma = \Sigma \Pi_{\mathcal{U}}^{(\Sigma)}$, so it is the $\Sigma$-orthogonal projection onto $\mathcal{U}$.

**Step 3.** At any minimizer $U^*$ we have

$$W - PU^* = W - \Pi_{\mathcal{U}}^{(\Sigma)} W = \big(I - \Pi_{\mathcal{U}}^{(\Sigma)}\big) W.$$

Since $\Pi_{\mathcal{U}}^{(\Sigma)}$ is a $\Sigma$-orthogonal projector,

$$\|W\|_\Sigma^2 = \big\|\Pi_{\mathcal{U}}^{(\Sigma)} W\big\|_\Sigma^2 + \big\|\big(I - \Pi_{\mathcal{U}}^{(\Sigma)}\big) W\big\|_\Sigma^2,$$

and the cross term is zero in the $\Sigma$ inner product. Therefore, the minimum is

$$\min_U \ \epsilon_{\mathrm{compress}}(U) = \big\|W - PU^*\big\|_\Sigma^2 = \big\|\big(I - \Pi_{\mathcal{U}}^{(\Sigma)}\big) W\big\|_\Sigma^2.$$

This proves the proposition with $U^* = \big(P^\top \Sigma P\big)^\dagger P^\top \Sigma W$ and $\epsilon_{\mathrm{compress}}(U^*) = \big\|\big(I - \Pi_{\mathcal{U}}^{(\Sigma)}\big) W\big\|_\Sigma^2$.

**Remarks.** (1) Nonzero mean: if $z$ has mean $\mu \neq 0$ and classifiers include biases, the optimal biases of the low-dimensional classifier absorb $\big(W^\top - (PU)^\top\big)\mu$, reducing the matrix part to the centered case with covariance $\Sigma = \mathrm{Cov}(z)$. (2) Uniqueness: if $P^\top \Sigma P \succ 0$ then $U^*$ is unique; if rank-deficient, $U^*$ may not be unique but $PU^*$ and the minimum value are unique.

## A.3 EXPERIMENTAL SETTINGS

Our experiments are conducted on a workstation with an NVIDIA RTX4090 GPU (24GB), Intel i9-10900KF CPU, and 32GB RAM, using PyTorch 2.4.1 with CUDA 12.4 and Python 3.10. We adopt the Adam optimizer (Kingma & Ba, 2014). For **Human-C**, **Human-D**, **Monkey-R**, **BCIC-2A**, and **BCIC-2B**, IND is configured with 32 embedding dimensions, 128 FFN dimensions, and 2 attention layers. For **FALCON-M1**, it uses 8 embedding dimensions and 32 FFN dimensions. For EEGConformer, ATCNet, CTNet, and EEGNet, we follow the Braindecode library implementation (Schirrmeister et al., 2017). For CWT+RNN, we use a 2-layer LSTM with 16 hidden dimensions. For CWT+CNN, we use a 2-layer 1D CNN, with 32 and 16 output channels, respectively, and a kernel size of 3. For IND (Spec), we replace spectrogram with CWT features, and compute 6 different spectral band of $\{[0\mathrm{Hz}, 7\mathrm{Hz}], [7\mathrm{Hz}, 13\mathrm{Hz}], [13\mathrm{Hz}, 30\mathrm{Hz}], [30\mathrm{Hz}, 70\mathrm{Hz}], [70\mathrm{Hz}, 150\mathrm{Hz}], [150\mathrm{Hz}, 400\mathrm{Hz}]\}$ and average the output into 10 time bins. For IND (STFT), we compute STFT with a window length of 128 and hop length of 64. For KD, VkD, TSKD (CE), we set the weight of distillation loss and task loss both to 0.5. The $\lambda$ of TSKD is by default 1, and is set to 0.1 on **FALCON-M1**.

## A.4 TASK-SPECIFIC RATIO AND CORRELATION COMPUTATION

For each session, we have teacher embeddings $\mathcal{Z}_\mathcal{T} \in \mathbb{R}^{d_t}$ of its test set, the projection matrix $P \in \mathbb{R}^{d_t \times d_s}$ and teacher classifier matrix $W_\mathcal{T} \in \mathbb{R}^{d_t \times K}$. For TSR, we compute based on the definition in equation 5. We estimate the mutual information between $\mathcal{Z}_\mathcal{T}$ and $P^\top \mathcal{Z}_\mathcal{T}$ using Canonical Correlation Analysis (CCA). CCA computes a set of canonical correlation coefficients $\{\rho_i(P)\}_{i=1}^k$, where $k = \min(d_t, d_s)$ and each $\rho_i(P)$ measures the maximal correlation between one linear projection of $\mathcal{Z}_\mathcal{T}$ and one linear projection of $P^\top \mathcal{Z}_\mathcal{T}$. The mutual information between the original and

projected teacher embeddings admits the closed-form expression

$$I\big(\mathcal{Z}_\mathcal{T}; P^\top \mathcal{Z}_\mathcal{T}\big) = -\frac{1}{2}\sum_{i=1}^{k}\log\big(1 - \rho_i(P)^2\big),\tag{8}$$

where larger values indicate that the projection $P$ preserves more information from the original teacher embeddings. We use a relative reconstruction which is then defined as:

$$\text{Relative Reconstruction Error} = \frac{\big\|\mathcal{Z}_\mathcal{T} - P^\top \mathcal{Z}_\mathcal{T} P\big\|_F^2}{\|\mathcal{Z}_\mathcal{T}\|_F^2},\tag{9}$$

where $\|\cdot\|_F$ denotes the Frobenius norm. This metric ranges in $[0, 1]$ and captures the proportion of information lost due to projection. The task performances and projection metrics are presented in Table 8.

Table 8: Task performances and projection metrics of three projection methods across three sessions on **Human-C**. MI represents mutual information, and RECON represents relative reconstruction error.

| SESSION 4-4 | | | | | |
|---|---|---|---|---|---|
| PROJECTIONS | TSR | MI | RECON | F1 | MACRO AVG |
| **TSKD** | **0.9374** | **106.17** | **0.00177** | **0.7318** | **0.5785** |
| TSKD (PCA) | 0.7832 | 156.49 | 0.00102 | 0.7166 | 0.5605 |
| TSKD (Random) | 0.7058 | 113.32 | 0.00174 | 0.7126 | 0.5449 |

| SESSION 4-5 | | | | | |
|---|---|---|---|---|---|
| PROJECTIONS | TSR | MI | RECON | F1 | MACRO AVG |
| **TSKD** | **0.9359** | **106.31** | **0.00180** | **0.7502** | **0.6143** |
| TSKD (PCA) | 0.7489 | 154.58 | 0.001226 | 0.7285 | 0.5751 |
| TSKD (Random) | 0.6943 | 113.34 | 0.001776 | 0.7224 | 0.5473 |

| SESSION 4-6 | | | | | |
|---|---|---|---|---|---|
| PROJECTIONS | TSR | MI | RECON | F1 | MACRO AVG |
| **TSKD** | **0.9340** | **106.698** | **0.00190** | **0.7716** | **0.6027** |
| TSKD (PCA) | 0.7525 | 157.1776 | 0.00110 | 0.7528 | 0.5826 |
| TSKD (Random) | 0.6837 | 113.866 | 0.00180 | 0.7576 | 0.5498 |

## A.5 DATASETS

### A.5.1 HUMAN-C

**Human-C** is a dataset for multi-class upperlimb movement classification obtained in the scope of a clinical trial aiming to restore upperlimb motor function after spinal cord injury. The participant was implanted ECoG (Mestais et al., 2014) and performs movement attempts involving hands, wrists, elbows and shoulders. Raw ECoG signals were initially sampled at 586Hz over 32 channels simultaneously. Signals were resampled at 500Hz and preprocessed as 1.5s windows with 100ms strides. The label is created based on the movement type of the last 0.1 second within each window. Each window is converted to 10 tokens after CWT tokenization.

We present the categories and number of samples of each session in **Human-C** in Table 9. In total we cover 6 different movement states, including rest, shoulder flexion, elbow extension, forearm pronation and hand opening and hand closing, but they differ across sessions. It is clear that the rest state always has the largest samples, and different movements are also not balanced. This implies the difficulty of directly train from scratch on this dataset. We extract 10 different wavelet feature, based on the following center frequencies: $\{10\text{Hz}, 30\text{Hz}, 50\text{Hz}, 60\text{Hz}, 70\text{Hz}, 80\text{Hz}, 90\text{Hz}, 120\text{Hz}, 150\text{Hz}, 200\text{Hz}\}$. All the models are trained with $\text{lr} = 3e-4$ and weight_decay $= 1e-4$.

In Table 10, we present results of distilling with TSKD on ATCNet, CTNet and EEGConformer. These models are selected as they share a transformer backbone, similar to the teacher model. Re-

sults show that TSKD consistently improve decoding performance on all baseline models, but these distilled baseline models still have worse performance than IND.

Table 9: Sample numbers of different categories in **Human-C**, each sample represents a 1.5 seconds neural recording.

| SETTING | SESSION | REST | SHOULDER FLEXION | ELBOW EXTENSION | WRIST PRONATION | HAND OPEN | HAND CLOSE |
|---|---|---|---|---|---|---|---|
| Train | 1 | 2808 | | 762 | 598 | 388 | |
| | 4 | 5487 | 388 | 927 | 796 | 705 | 921 |
| | 16 | 14130 | | 1105 | | 877 | 788 |
| Test | 1 | 776 | | 193 | 178 | 197 | |
| | 4 | 1380 | 211 | 214 | 226 | 186 | 195 |
| | 5 | 1260 | 158 | 244 | 214 | 200 | 193 |
| | 6 | 1812 | 144 | 176 | 173 | 226 | 151 |
| | 17 | 1292 | | 125 | | 188 | 95 |

Table 10: Movement decoding performance on **Human-C**. The number of parameters are indicated for each model. For each split, we report the mean and standard deviation of F1 and Avg Recall over three random initializations. Models are selected based on a validation set comprising 20% of the training data. Scores are reported in %. Our method and the best results are shown in **bold**, with the second-best results underlined.

| SESSION | 1-1 | | 4-4 | | 4-5 | | 4-6 | | 16-17 | |
|---|---|---|---|---|---|---|---|---|---|---|
| METRICS | F1 | Avg Recall | F1 | Avg Recall | F1 | Avg Recall | F1 | Avg Recall | F1 | Avg Recall |
| Teacher (100M) | 77.6 | 67.4 | 73.3 | 57.2 | 74.5 | 60.8 | 77.3 | 62.5 | 80.6 | 58.9 |
| Conformer (630K) | 37.7 ±11.1 | 38.3 ±4.36 | 40.8 ±5.21 | 31.5 ±2.79 | 36.0 ±8.45 | 33.9 ±13.9 | 39.9 ±6.39 | 23.7 ±1.61 | 43.1 ±7.58 | 41.4 ±5.05 |
| ATCNet (114K) | 44.9 ±3.80 | 33.9 ±3.90 | 48.4 ±5.39 | 26.9 ±3.37 | 45.2 ±1.94 | 22.4 ±2.59 | 54.4 ±3.95 | 23.2 ±2.03 | 43.7 ±3.10 | 53.8 ±1.29 |
| CTNet (152K) | 46.1 ±0.40 | 30.5 ±1.11 | 49.1 ±2.25 | 27.2 ±1.57 | 42.2 ±2.25 | 23.5 ±1.45 | 54.3 ±0.77 | 21.6 ±2.04 | 44.3 ±3.20 | 49.7 ±3.29 |
| EEGNet (5.5K) | 29.0 ±16.8 | 39.2 ±4.57 | 42.9 ±1.89 | 28.8 ±3.75 | 41.8 ±4.76 | 25.1 ±7.98 | 42.8 ±6.38 | 17.5 ±1.59 | 54.1 ±9.01 | 43.7 ±6.43 |
| LaBraM (5.8M) | 44.9 ±5.35 | 28.1 ±1.85 | 57.8 ±0.95 | 34.3 ±2.14 | 56.9 ±1.02 | 34.4 ±1.24 | 60.9 ±0.77 | 25.9 ±1.18 | 71.9 ±2.21 | 37.9 ±2.89 |
| **IND (30K)** | 69.1 ±1.64 | 56.6 ±4.11 | 64.9 ±3.45 | 44.9 ±3.78 | 61.5 ±2.20 | 37.1 ±5.68 | 70.6 ±0.97 | 44.2 ±5.26 | 72.3 ±0.83 | 52.1 ±5.77 |
| Conformer + TSKD | 53.4 ±0.78 | 48.2 ±3.79 | 63.2 ±4.48 | 47.6 ±3.85 | 63.6 ±5.69 | 46.7 ±5.28 | 65.7 ±2.08 | 42.2 ±1.49 | 47.3 ±19.9 | 50.7 ±8.15 |
| ATCNet + TSKD | 51.8 ±2.27 | 41.2 ±2.61 | 64.8 ±2.07 | 47.1 ±1.89 | 61.1 ±2.93 | 36.9 ±4.58 | 66.6 ±2.08 | 36.2 ±4.58 | 49.6 ±13.4 | 40.1 ±5.84 |
| CTNet + TSKD | 53.2 ±15.4 | 46.5 ±4.77 | 59.8 ±2.32 | 40.4 ±4.51 | 55.9 ±1.79 | 33.6 ±2.54 | 63.7 ±0.69 | 30.3 ±0.43 | 61.6 ±2.10 | 50.8 ±9.04 |

### A.5.2 MONKEY-R

**Monkey-R** is a dataset for continuous upperlimb joint trajectory regression. A macaque was implanted with 2 of 32 channels of ECoG, bilaterally on the sensorimotor cortex. It performs reach and pull tasks. Each time the monkey tries to grasp a cylinder on a multiaxis robotic arm. A successful trial is recognized and validated when the arm is pulled towards the animal and crosses a defined distance. 5 to 14 markers are positioned on the arm and hand for the 3D kinematic tracking of the movement. The x-coordinate of the monkey's wrist position is used for our application. The ECoG is recorded at 2000 Hz and was resampled to 500 Hz. Similar to **Human-C**, we also create 1.5 seconds segments with 0.1-second stride, and the target is the average x-coordinate of the wrist position in the last 0.25 second of the segment. Each window is converted to 10 tokens after CWT tokenization.

Here presents number of samples and $R^2$ values of different models across sessions. day0 represents the validation set, each sample is a 1.5 seconds recording, and in training set we have 35705 samples. We extract 5 different wavelet feature, based on the following center frequencies: $\{10\text{Hz}, 30\text{Hz}, 60\text{Hz}, 80\text{Hz}, 100\text{Hz}\}$. All the model is trained with $\text{lr} = 3e - 3$ and weight_decay $= 1e - 4$.

**Online Decoding Experiment** We conducted a cross-session online decoding experiment on day 5 and present the recorded real-time ground truth and decoding output in the supplementary material. During the task, the monkey's ECoG signals were collected and transmitted to an NVIDIA Jetson device via the Lab Streaming Layer (Kothe et al., 2025), where the weights of IND were stored. IND generated decoding outputs from the streaming ECoG signals, and a custom UI was developed to visualize the results. As shown in Figure 4, the left panel displays trajectories: the top row shows the actual trajectory, and the bottom row shows the prediction. Each window visualizes the most recent 10 seconds. In the middle panel, the red bar indicates the decoding output at the current time

Table 11: Decoding performance across different days on **Monkey-R**. The $R^2$ and the number of samples of each session are reported. Our method name and the best performances are **bolded**, and the second best are underlined.

| DATES | SAMPLES | **IND** | CWT+RNN | CWT+CNN | IND (SPEC) | IND (STFT) | EEGCONFORMER | ATCNET | CTNET | EEGNET | LABRAM |
|---|---|---|---|---|---|---|---|---|---|---|---|
| day0 | 8926 | **0.7533** | 0.5985 | 0.5543 | 0.6105 | 0.6088 | 0.7277 | 0.6294 | 0.7305 | 0.5707 | 0.5104 |
| day1 | 8466 | **0.5643** | 0.4080 | -0.4222 | -0.2297 | -0.9583 | -0.1053 | 0.1369 | 0.3823 | 0.1147 | 0.1734 |
| day5 | 15174 | **0.6835** | 0.5511 | -0.4498 | 0.5195 | -2.276 | -0.3824 | -0.0579 | 0.3815 | 0.0964 | 0.3224 |
| day6 | 17144 | **0.6728** | 0.3777 | 0.3202 | 0.6306 | -0.2092 | -0.1716 | 0.0689 | 0.4748 | 0.3093 | 0.3 |
| day8 | 7897 | **0.7155** | 0.3881 | 0.6155 | 0.5915 | 0.2819 | 0.6025 | 0.5337 | 0.6038 | 0.5051 | 0.1918 |
| day12 | 17075 | **0.3630** | 0.3008 | 0.2018 | 0.3383 | -0.3433 | -1.4514 | -1.5261 | 0.1074 | -0.0930 | 0.2469 |
| day14 | 17740 | **0.4479** | 0.3069 | 0.3649 | 0.1694 | -0.0362 | -0.0124 | -0.1029 | 0.0691 | 0.3967 | 0.1943 |
| day15 | 17331 | 0.5448 | 0.3342 | **0.6057** | 0.2222 | -0.0542 | 0.2580 | 0.2335 | 0.1607 | 0.4509 | 0.153 |

point. The right panel illustrates one of the five wavelet features across 64 channels at the same time point, where red denotes higher values. We present a one-minute recording of the online decoding test, which demonstrates that IND achieves strong performance with low latency.

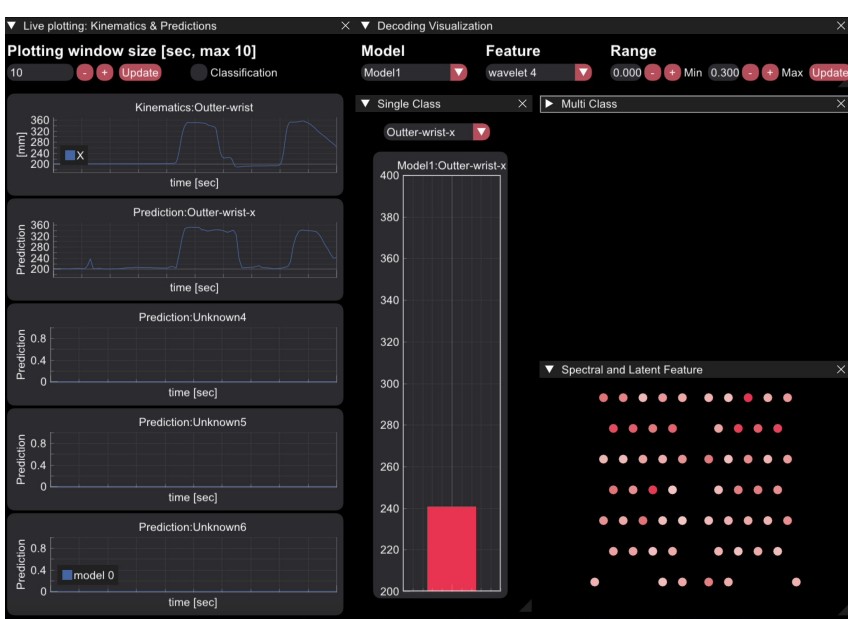

Figure 4: Display of the user interface to visualize online decoding results.

### A.5.3 HUMAN-D

Each sample from **Human-D** is a segment of 64 channels and 2 seconds recording of 500Hz. Therefore, we create 10 tokens, each represents 0.2 second, and extract 7 wavelet features: $\{10\text{Hz}, 20\text{Hz}, 40\text{Hz}, 60\text{Hz}, 80\text{Hz}, 100\text{Hz}, 125\text{Hz}\}$. All of the models are trained for 30 epochs with lr $= 3e - 3$. The sample numbers of the training set and test set of **Human-D** are listed in Table 12. We perform ablation study of IND on **Human-D** as well, where its performance is compared with CWT+RNN, CWT+CNN, IND (Spec) and IND (STFT). According to Table 13, IND and IND (STFT) have the best performance, where IND performs better in inter-subject decoding.

### A.5.4 BCIC-2A AND BCIC-2B

For the two EEG datasets, we change the tokenization approach of our decoder in order to match EEGPT. Given an EEG sample $x \in \mathbb{R}^{C \times 1024}$, where 1024 represents 4 seconds with sample rates of 256 Hz, $C = 22$ for **BCIC-2A** and $C = 3$ for **BCIC-2B**, we directly treat each channel as a token. Table 14 shows the sample distributions of different splits. Teacher model EEGPT is fine-tuned on large $\mathcal{X}_{\text{offline}}$, and distillation is performed on a much smaller and imbalanced $\mathcal{X}_{\text{recalib}}$. Distilled model is trained with lr $= 4e - 4$ and OneCycleLR scheduler.

We perform distillation experiments on these two datasets with EEGConformer and present results in Table 15. According to the results, all distillation methods cannot significantly improve per-

Table 12: Sample numbers of different categories in training and testing set of **Human-D**, each sample represents a 2 seconds neural recording.

| SPLITS | SUBJECTS | ARM REST | ARM MOVE |
|---|---|---|---|
| | 1 | 4986 | 5049 |
| | 2 | 5387 | 5368 |
| | 3 | 5082 | 5073 |
| | 4 | 5344 | 5316 |
| | 5 | 5399 | 5365 |
| | 6 | 4844 | 4755 |
| Train | 7 | 5964 | 4945 |
| | 8 | 4921 | 4808 |
| | 9 | 5412 | 5436 |
| | 10 | 5012 | 5082 |
| | 11 | 4844 | 4812 |
| | 12 | 4796 | 4778 |
| | 1 | 186 | 186 |
| | 2 | 45 | 45 |
| | 3 | 102 | 102 |
| | 4 | 115 | 115 |
| | 5 | 129 | 129 |
| Test | 6 | 180 | 180 |
| | 7 | 173 | 173 |
| | 8 | 178 | 178 |
| | 9 | 21 | 21 |
| | 10 | 154 | 154 |
| | 11 | 184 | 184 |
| | 12 | 191 | 191 |

Table 13: Ablation experiment result for Movement detection performance on **Human-D**. The average accuracy and AUROC over 12 patients are reported. Scores are in %. Our method name and the best performances are **bolded**, and the second best are underlined.

| SETTING | METRIC | **IND** | CWT+RNN | CWT+CNN | IND (SPEC) | IND (STFT) |
|---|---|---|---|---|---|---|
| Intra-Subject | Acc | 77.1 | 53.5 | 73.4 | 64.5 | **78.6** |
| | AUROC | 85.0 | 61.5 | 84.2 | 80.9 | **86.6** |
| Inter-Subject | Acc | 54.5 | 48.3 | **55.5** | 50.8 | 54.3 |
| | AUROC | **59.8** | 48.1 | 55.4 | 55.8 | 57.1 |

formances on BCIC2A, but improve performances on BCIC2B greatly. TSKD has a comparable performance, but doesn't show a significant advantage. This is expected as EEGConformer has a much larger size than IND, thus the capacity gap between student and teacher is smaller, leading to weaker improvement for task-specific distillation.

Table 14: Sample numbers of different categories in different splits of **BCIC-2A** and **BCIC-2B**, each sample represents a 4 seconds neural recording.

| DATASET | SPLIT | LEFT HAND | RIGHT HAND | FEET | TONGUE |
|---|---|---|---|---|---|
| BCIC2A | $\mathcal{X}_{\text{offline}}$ | 1034 | 1036 | 1038 | 1036 |
| | $\mathcal{X}_{\text{recalib}}$ | 11 | 34 | 57 | 116 |
| | Test | 144 | 144 | 144 | 144 |
| BCIC2B | $\mathcal{X}_{\text{offline}}$ | 2610 | 2610 | | |
| | $\mathcal{X}_{\text{recalib}}$ | 290 | 87 | | |
| | Test | 360 | 360 | | |

### A.5.5 **FALCON-M1**

These two datasets are collected from FALCON benchmark (Karpowicz et al., 2024). **FALCON-M1** uses the held-in data of M1-A from FALCON, which consists of recordings using Floating Microelectrode Arrays (Microprobes), implanted in the precentral gyrus while the monkey reached to, grasped, and manipulated an object in a specific location (4 possible objects, 8 possible locations). Intramuscular electromyography (EMG) was recorded from 16 muscles in the right hand and upper extremity as the regression target. M1-A consists of 4 held-in datasets spanning 5 days, each with 53-61 minutes of calibration data. For distillation, we use 1% of training data for each day as $\mathcal{X}_{\text{recalib}}$, and report the average results on calibration data of 4 days. All of the distillation method consists of a distillation loss plus the MSE-based regression loss.

Table 15: Distilled performance for EEGConformer on two EEG datasets . For **BCIC-2A**, the average weighted F1 over 9 subjects is reported; for **BCIC-2B**, the average AUROC over 9 subjects is reported. We report the mean and std of scores over 3 random initializations. Scores are in %. Our method name and the best performances are **bolded**, and the second best are underlined.

| DATASET | TEACHER | CONFORMER | +KD | +SIMKD | +VKD | +RDIMKD | +TOFD | +TED | +TSKD | +TSKD(CE) |
|---------|---------|-----------|-----|--------|------|---------|-------|------|-------|-----------|
| BCIC2A | 50.66 | 29.64 | 29.45 ±3.83 | 24.15 ±3.03 | **30.11 ±3.87** | 24.85 ±2.74 | 28.74 ±4.04 | 29.01 ±3.53 | 29.56 ±4.28 | 29.20 ±4.08 |
| BCIC2B | 80.51 | 68.44 | 74.00 ±0.37 | 65.30 ±0.29 | 75.68 ±0.41 | 72.90 ±0.73 | 75.67 ±1.09 | **77.77 ±0.77** | 76.76 ±0.93 | 76.86 ±0.91 |

## A.6  QUANTIZATION AND POWER ESTIMATION

### A.6.1  QUANTIZATION AWARE TRAINING

For all the weights quantization in the decoder, we apply channel-wise scaler by estimating the range of weight matrix, like in I-ViT (Li & Gu, 2023). For activations, we estimate a fixed range for non-essential operations, and apply PACT (Choi et al., 2018) on value-sensitive operations. Table 16 illustrates the details of clipping range in each operation.

Table 16: Clipping ranges of different activations in the quantized decoder. $max(|\text{min}|, |\text{max}|)$ means the matrix range is estimated to be the clipping range, and other values mean that $\alpha$ is initialized and it is updated during QAT.

| OPERATION | CLIPPING RANGE | OPERATION | CLIPPING RANGE |
|-----------|----------------|-----------|----------------|
| Wavelet | 1.0 | Layer 2 Q projection | 7.0 |
| Input Layer | $max(|\text{min}|, |\text{max}|)$ | Layer 2 K projection | 5.0 |
| Positional Encoding | $max(|\text{min}|, |\text{max}|)$ | Layer 2 V projection | 5.0 |
| Layer 1 Q projection | 1.5 | Layer 2 QK multiplication | 20.0 |
| Layer 1 K projection | 2.0 | Layer 2 QK normalizer | 150.0 |
| Layer 1 V projection | 1.5 | Layer 2 Attention output | 100.0 |
| Layer 1 QK multiplication | 3.0 | Layer 2 Out Projection | 1.5 |
| Layer 1 QK normalizer | 25.0 | Layer 2 Residual & LayerNorm | $max(|\text{min}|, |\text{max}|)$ |
| Layer 1 Attention output | 10.0 | Layer 2 Feedforward Layer | $max(|\text{min}|, |\text{max}|)$ |
| Layer 1 Out Projection | 1.0 | Classifier | $max(|\text{min}|, |\text{max}|)$ |
| Layer 1 Residual & LayerNorm | $max(|\text{min}|, |\text{max}|)$ | | |
| Layer 1 Feedforward Layer | $max(|\text{min}|, |\text{max}|)$ | | |

In order to compute linear attention, we perform a quantized division with inter-only operations. For two quantized matrix $A$ and $B$ and their scales $s_a$ and $s_b$, the quantized divison generates $O * s_o \approx \frac{A*s_a}{B*s_b}$, where $O * s_o$ is an approximation of the float point result of the division. We first obtain $s_o$ from the output range of the full-precision model, and then $O$ only need to be computed from $\frac{A*s_a}{B*s_b*s_o}$. Dividing two positive integer without remainder will leave quantization error up to 1, and when the numerator is smaller, the result is 0 which is meaningless. In order to shrink this error, we perform the following division:

$$O \approx \frac{A * s_a * 2^e}{B * s_b * s_o} \gg e. \tag{10}$$

Scaling numerator with $2^e$ ensures it is larger than the denominator, and also reduces the quantization error of each element to lower than $2^{-e}$. In practice we select $e = 12$.

### A.6.2  POWER ESTIMATION FOR ON CHIP IMPLEMENTATION

We estimate power based on 45nm techniques according to the parameters in Horowitz (2014). We first compute the energy cost for each computation, leakage and I/O, estimate power based on the frequency of computing wavelet, which is 10Hz as each 100ms an averaged wavelet feature can be computed. Table 17 shows that leakage power dominates the energy cost, especially when the chip is working with a relatively low frequency. Therefore, minimizing the memory cost of model weights is the most important for the implementation.

## A.7  WAVELET FEATURE EXTRACTION

### A.7.1  COMPARING WAVELET AND CONVOLUTION FEATURES

We visualize the feature computed by wavelet from our decoder and compare it with the feature learned from the convolution block in CTNet on subject 4 of **Human-D** with intra-session setting.

Table 17: Power analysis based on 45nm techeniques, the total power consists of operation power, leakage power, and I/O power. The power represents the energy cost for one computation cycle, which is 50ms.

| PRECISION | OPERATION POWER (J) | LEAKAGE POWER (J) | I/O POWER (J) | TOTAL POWER (J) |
|---|---|---|---|---|
| FP | $1.682 \times 10^{-6}$ | $1.130 \times 10^{-3}$ | $1.069 \times 10^{-5}$ | $1.142 \times 10^{-3}$ |
| W8A8 | $8.409 \times 10^{-8}$ | $2.824 \times 10^{-4}$ | $5.347 \times 10^{-7}$ | $2.83 \times 10^{-4}$ |

Figure 5: Wavelet feature of subject 4 from **Human-D**. The training set and test set are projected to two two-dimensional space via UMAP respectively.

Figure 5 and Figure 6 shows that the convolution block of CTNet cannot effectively extract the discriminative information from ECoG signals. On the contrary, only using a wavelet without any learnable parameters already extracts separable features.

### A.7.2 WAVELET FREQUENCY SELECTION

Table 18: Intra-session decoding performance on **Human-D**. The average accuracy and AUROC over 12 patients are reported. Default decoder extracts 7 wavelet features: $\{10\text{Hz}, 20\text{Hz}, 40\text{Hz}, 60\text{Hz}, 80\text{Hz}, 100\text{Hz}, 125\text{Hz}\}$, others remove two frequencies from the 7 original frequencies. For example, / $\{10\text{Hz}, 20\text{Hz}\}$ only extract 5 wavelet frequencies: $\{40\text{Hz}, 60\text{Hz}, 80\text{Hz}, 100\text{Hz}, 125\text{Hz}\}$. Scores are in %.

| METRIC | DEFAULT | / {10Hz, 20Hz} | / {20Hz, 40Hz} | / {40Hz, 60Hz} | / {60Hz, 80Hz} | / {80Hz, 100Hz} | / {100Hz, 125Hz} |
|---|---|---|---|---|---|---|---|
| Acc | **77.1** | 74.5 | 74.7 | 76.0 | 75.7 | 75.9 | 76.1 |
| AUROC | **85.0** | 85.2 | 82.4 | 83.7 | 83.9 | 84.1 | 84.3 |

We study which frequency bands are more important for the movement decoding task when we perform wavelet feature extraction. We perform an ablation study on **Monkey-R** and **Human-D**, where a certain frequency is removed, and we compare the performance with the full frequency. In Table 18, we can see that all the frequency bands from 10Hz to 125Hz are important for detecting movements.

On **Monkey-R**, we test the long-term decoding performance with different wavelet frequencies, which is illustrated in Figure 7. The result shows that all decoders have the same trend across the sessions, thus the drifting of neural signals occurs on all frequency bands. Furthermore, 30Hz is really essential in the task, as removing it will reduce a lot of performance. High frequencies from 60Hz to 100Hz does not affect the performance much, and even hurt the generalization in a longer time.

Our experimental observations are consistent with previous studies showing that different frequency bands in motor-related ECoG activity are associated with distinct functional roles (Jiang et al., 2020; Volkova et al., 2019; Tam et al., 2019). The beta band (13–30 Hz) is strongly modulated around movement onset and termination, typically showing suppression during preparation and execution followed by a post-movement rebound, thereby serving as a reliable indicator of motor state transitions. The gamma band (30–80 Hz) is predominantly engaged during movement execution, with robust increases in power observed during grasping and fine finger movements, reflecting its close link to active motor output. The high-gamma and broadband high-frequency range ($> 80$ Hz, e.g., 100–200 Hz or higher) provides the most precise information about muscle activity, movement kinematics, and force dynamics. This aligns with our results showing that activity around 30 Hz is important for detecting movement onset and termination, thereby influencing trajectory regression

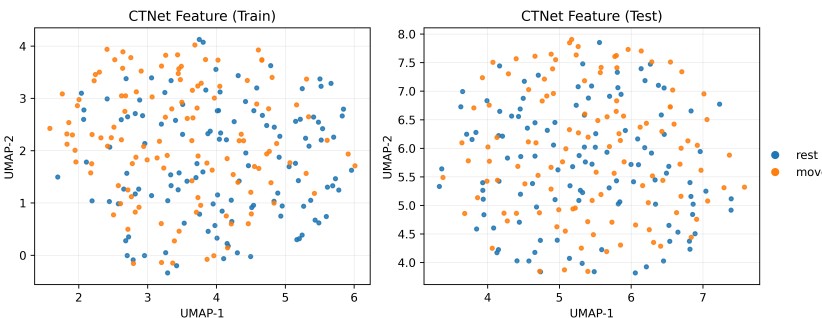

Figure 6: Convolution feature of subject 4 from **Human-D**. The training set and test set are projected to two dimensional space via UMAP respectively.

performance, while broadband frequencies are broadly involved in capturing general motor activities.

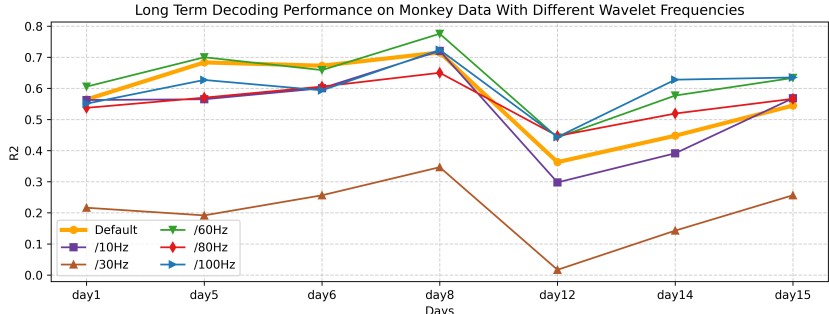

Figure 7: Long term decoding performance on **Monkey-R** with different frequencies. Default decoder uses 5 frequencies: $\{10\text{Hz}, 30\text{Hz}, 60\text{Hz}, 80\text{Hz}, 100\text{Hz}\}$, and other five decoders remove 1 of 5 frequencies.

