# OpenReview forum: "BrainDistill: Implantable Motor Decoding with Task-Specific Knowledge Distillation"
_ICLR.cc/2026/Conference — Submitted to ICLR 2026_

### Official Review · Reviewer_JvuC · 2025-10-23

**Soundness:** 2
**Presentation:** 2
**Contribution:** 3
**Rating:** 4
**Confidence:** 4

**Summary:**

Braindistill: implantable motor decoding with task-specific knowledge distillation presents three main contributions: a linear attention Transformer (IND), a distillation technique (TKSD), and metric to evaluate the data-specific effectiveness of a projection (TSR). This work provides compelling empirical evidence for all three components, albeit mostly on private datasets.

**Strengths:**

1. Advantage over the state-of-the-art on the datasets considered is clear
2. Ablations are thorough and validate the choices of the authors
3. The stability shown under quantization is a nice bonus

**Weaknesses:**

1. While the paper advertises 6 datasets (3 private and 3 public), the relevant performance comparisons against baseline models are only performed on the 3 private ones, and the public ones are left for ablations. The same benchmarking must be performed on the public ones to ensure unbiased results.
2. A lighter version of EEGPT (the teacher model) needs to be included as a baseline. As it stands, it is not clear whether EEGPT performs better than IND because of the size or because of the architecture.
3. Related to the point above, I assume that EEGPT is chosen due to its large size (100M), while the others are considerably smaller. This should be made clear, and the parameters of all models also need to be reported.
4. The paper is quite disjointed and tough to parse. I found it rather difficult to read and decipher how all the components go together.

**Questions:**

1. It’s not fully clear what “training from scratch” means for the student model. For distillation, from my understanding the teacher is trained on 159 sessions (X_{offline}), then the student is distilled using X_{recalib}, then the student is tested on X_{online}. For the “scratch” case, is the student trained on X_{offline}, X_{recalib}, or both at the same time?
2. The definition of X_{offline} is not fully clear. Is it patient-specific or not?
3. \Delta is not defined in Eq. 1
4. What’s the advantage of not using the same classifier during the two phases of distillation?
5. The division of the splits is quite confusing at first, should make clear that, e.g., 1-1 means training on the training split of session 1 and testing on the testing split of session 1.
6. The second term of L_{TKSD} is highly reminiscent of ridge regression, you might find works on the Tikhonov factor useful for the determination of \lambda.
7. For clarity, it should be specified that the classifier must be a single layer + non-linearity

---

> ### Author Response · Authors · 2025-11-19
> **Answer 1 (1/2)**
>
> We sincerely thank the reviewer for appreciating the empirical advantages of BrainDistill and for the detailed suggestions and guidance regarding the presentation. We have improved the manuscript accordingly, and we address the raised concerns in detail below.
>
> ## **1. (1/2) Experiment coverage on private and public datasets:**
>
> **Weakness 1: While the paper advertises 6 datasets (3 private and 3 public), the relevant performance comparisons against baseline models are only performed on the 3 private ones, and the public ones are left for ablations. The same benchmarking must be performed on the public ones to ensure unbiased results.**
>
> We would like to kindly clarify that, **as shown in Table 1, our study includes four public datasets (Human-D, BCIC-2A, BCIC-2B, FALCON-M1) and two private datasets (Human-C, Monkey-R). Both TSKD and IND are benchmarked against strong baselines on these public and private datasets. Specifically, TSKD is evaluated on Human-C, BCIC-2A, BCIC-2B, and FALCON-M1, while IND is evaluated on Human-C, Human-D, and Monkey-R.** The results indicate that IND achieves superior performances on both private and public ECoG datasets, and that TSKD performs well across modalities on both private and public datasets. We perform quantization experiments on Human-C because the model is designed to be deployed based on this clinical setting.
>
> To ensure that our methods are thoroughly and transparently benchmarked, we further conducted an ablation study of IND on Human-D , as well as additional distillation experiments with EEGConformer on BCIC-2A and BCIC-2B.
>
> Here are the ablation results on Human-D, which show IND and IND (STFT) have the best performance, and IND has better cross-subject accuracy. These results validate the design of IND.
>
> | Setting        | Metric | IND | CWT+RNN | CWT+CNN | IND (Spec) | IND (STFT) |
> |----------------|--------|------|---------|---------|-------------|-------------|
> | Intra-Subject  | Acc   | 77.1 | 53.5 | 73.4 | 64.5 | **78.6** |
> | Intra-Subject  | AUROC | 85.0 | 61.5 | 84.2 | 80.9 | **86.6** |
> | Inter-Subject  | Acc   | 54.5 | 48.3 | **55.5** | 50.8 | 54.3 |
> | Inter-Subject  | AUROC | **59.8** | 48.1 | 55.4 | 55.8 | 57.1 |
>
> For the EEG datasets, we find that improving EEGConformer on BCIC-2A is particularly challenging, whereas results on BCIC-2B show substantial performance gains. Overall, TSKD remains competitive across model architectures and modalities, but it does not exhibit clear advantages over TOFD and TED in the EEG setting, likely because the capacity gap between EEGConformer and EEGPT is considerably smaller.
>
> | Dataset | Teacher | EEGConformer | +KD | +SimKD | +VkD | +RdimKD | +TOFD | +TED | **+TSKD** | **+TSKD(CE)** |
> |---------|---------|------|----------------|----------------|----------------|----------------|----------------|----------------|----------------|----------------|
> | **BCIC2A** | 50.66 | 29.64 | 29.45 ± 3.83 | 24.15 ± 3.03 | **30.11 ± 3.87** | 24.85 ± 2.74 | 28.74 ± 4.04 | 29.01 ± 3.53 | 29.56 ± 4.28 | 29.20 ± 4.08 |
> | **BCIC2B** | 80.51 | 68.44 | 74.00 ± 0.37 | 65.30 ± 0.29 | 75.68 ± 0.41 | 72.90 ± 0.73 | 75.67 ± 1.09 | **77.77 ± 0.77** | 76.76 ± 0.93 | 76.86 ± 0.91 |
>
> These analyses strengthen the evidence that our proposed methods generalize across public datasets and models.

---

> ### Author Response · Authors · 2025-11-19
> **Answer 1 (2/2)**
>
> ## **1. (2/2) Experiment coverage on private and public datasets:**
>
> Here we present an overview of all experiments and their corresponding datasets:
>
> | Study / Dataset             | Human-C (private) | Monkey-R (private) | Human-D (public) | BCIC-2A (public) | BCIC-2B (public) | FALCON-M1 (public) |
> |-----------------------------|--------------------|----------------------|-------------------|-------------------|-------------------|----------------------|
> | IND train from scratch      | √                  | √                    | √                 |                   |                   |                      |
> | IND distillation            | √                  |                      |                   | √                 | √                 | √                    |
> | EEGConformer distillation   | √                  |                      |                   | √                 | √                 |                      |
> | IND ablation study          |                    | √                    | √                 |                   |                   |                      |
> | IND quantization study      | √                  |                      |                   |                   |                   |                      |
> | TSR metrics evaluation      | √                  |                      |                   |                   |                   |                      |
>
>
> Nevertheless, there are important motivations for leveraging the private datasets in our benchmarking. As the data of Human-C is collected during a long period, and different movement tasks are performed across sessions, therefore our teacher classifier is also fine-tuned and evaluated across multiple sessions. Here shows the results of the fine-tuned teacher classifier and a classifier only trained with $\mathcal{X}_{\text{recalib}}$:
>
> | Model            | 1-1 F1 | 1-1 Avg Recall | 4-4 F1 | 4-4 Avg Recall | 4-5 F1 | 4-5 Avg Recall | 4-6 F1 | 4-6 Avg Recall | 16-17 F1 | 16-17 Avg Recall |
> |------------------|--------|---------|--------|---------|--------|---------|--------|---------|----------|-----------|
> | Teacher (classifier fine-tuned)   | 77.6   | 67.4    | 73.3   | 57.2    | 74.5   | 60.8    | 77.3   | 62.5    | 80.6     | 58.9      |
> | Teacher (classifier re-trained)   | 74.9   | 64.9    | 69.0   | 48.8    | 68.5   | 49.7    | 72.6   | 46.2    | 75.5     | 48.1      |
>
> Although the learned embeddings of the teacher model are the same and kept fixed, training a classifier only on limited data leads to a significant performance drop, especially for average recall. This illustrates the importance of leveraging task-specific information from a well-trained teacher classifier. Due to this reason, we chose to illustrate the TSR results on the private Human-C dataset because its teacher classifier contains the richest task-specific information. This setting therefore provides a better demonstration of how conventional projection methods and evaluation metrics fail to capture this task-specific knowledge.

---

> ### Author Response · Authors · 2025-11-19
> **Answer 2**
>
> ## **2. Question regarding teacher model performance and model parameters:**
>
> **Weakness 2: A lighter version of EEGPT (the teacher model) needs to be included as a baseline. As it stands, it is not clear whether EEGPT performs better than IND because of the size or because of the architecture.**
>
> **Weakness 3: Related to the point above, I assume that EEGPT is chosen due to its large size (100M), while the others are considerably smaller. This should be made clear, and the parameters of all models also need to be reported.**
>
> We would like to first point out that there are two teacher models with around 100M parameters. The teacher model (100M) used in Human-C is not EEGPT, but a private transformer model pre-trained with self-supervision on 159 sessions, whose embeddings and logits are provided together with the data, and the model architecture is not provided currently. The EEGPT (101M) is used for BCIC-2A and BCIC-2B, as these are the downstream datasets in its original settings. The teacher model on Human-C outperforms IND because it extracts more complex features, as it uses 24 wavelet frequencies, but IND only uses 10, and it is trained on more sessions. One proof for this is that if the teacher classifier is only trained on $X_{recalib}$, its performance significantly drops, as mentioned in Answer 1.
>
> In addition, we also evaluate linear probing on LaBraM as a baseline, and it performs poorly on both Human-C and Monkey-R. **These observations indicate that a well-trained teacher classifier is just as crucial as the model architecture and model size. This further motivates TSKD, whose goal is to preserve as much task-specific knowledge from the teacher model as possible.**
>
> We report all the model sizes in the experiment of Human-C:
>
> | Model |
> |-------|
> | Teacher (100M) |
> | Conformer (630K) |
> | ATCNet (114K) |
> | CTNet (152K) |
> | EEGNet (5.5K) |
> | LaBraM (5.8M) |
> | **IND (30K)** |
>
> The table shows that our baselines cover a wide range of sizes, from 5.5K to 5.8M, and IND still performs the best among them.

---

> ### Author Response · Authors · 2025-11-19
> **Answer 3**
>
> ## **3. Problems regarding the presentation of the manuscripts:**
>
> **Weakness 4: The paper is quite disjointed and tough to parse. I found it rather difficult to read and decipher how all the components go together.**
>
> We have improved the writing based on the reviewers’ comments and concerns. Our manuscript now follows a clearer and more coherent structure. First, we introduce the existing problem of the capacity gap between neural foundation models and implantable decoders. We then present four contributions aimed at addressing this challenge: task-specific knowledge distillation, the task-specific ratio metric, an implantable neural decoder, and an integer-only QAT method. In the experimental section, we benchmark each of these proposed components against strong baselines across six datasets. In order to give a clear intuition of the proposed TSKD method, we added a paragraph in section 2.2 explaining the difference between TSKD and conventional distillation methods. Besides, we added quantitative experiments on the TSR metric to illustrate its correlation with task performances.
>
> Here are the replies to the raised comments:
>
> ### Definition of training from scratch:
>
> **Question 1: It’s not fully clear what “training from scratch” means for the student model. For distillation, from my understanding the teacher is trained on 159 sessions (X_{offline}), then the student is distilled using X_{recalib}, then the student is tested on X_{online}. For the “scratch” case, is the student trained on X_{offline}, X_{recalib}, or both at the same time?**
>
> We have added an explanation, as “Training from scratch” means training the student model on $X_{recalib}$ without distillation. All the neural decoders only have access to $X_{recalib}$, and only the teacher model has access to $X_{offline}$.
>
> ### Definition of $X_{offline}$:
>
> **Question 2: The definition of X_{offline} is not fully clear. Is it patient-specific or not?**
>
> There is no limitation on whether $X_{offline}$ is patient-specific or not. For the teacher model of Human-C, $X_{offline}$ is the 159 sessions of the same patient, while for NDT2 and EEGPT, $X_{offline}$ is their pre-trained dataset, which includes multiple patients. We have benchmarked both scenarios to ensure that TSKD adapts to different teacher models.
>
> ### Definition of $\Delta$:
>
> **Question 3: \Delta is not defined in Eq. 1**
>
> Thanks for pointing out, \Delta refers to the approximation error between teacher embeddings and the projected embeddings. This represents that the projected embeddings are a low-rank approximation of the teacher embeddings. We have added the explanation in the revised manuscript.
>
> ### Notation of splits:
>
> **Question 5: The division of the splits is quite confusing at first, should make clear that, e.g., 1-1 means training on the training split of session 1 and testing on the testing split of session 1.**
>
> Thanks for the comment. A clear explanation of the splitting has been updated in the revised paper.
>
> ### Definition of classifier:
>
> **Question 7: For clarity, it should be specified that the classifier must be a single layer + non-linearity**
>
> We have specified in the revised section 2.1 that the classifier f in our task is simply one linear layer.

---

> ### Author Response · Authors · 2025-11-19
> **Answer 4, 5**
>
> ## **4. Using the same classifier in the two phases of distillation:**
>
> **Question 4: What’s the advantage of not using the same classifier during the two phases of distillation?**
>
> Thanks for the insightful question. We have tested using the same classifier or randomly initialize the student classifier during phase 2 of distillation on 4-4, 4-5 and 4-6 of Human-C, and the results show that the randomly initialized classifier performs better:
>
> | Model             | 4-4 F1 | 4-4 Avg Recall | 4-5 F1 | 4-5 Avg Recall | 4-6 F1 | 4-6 Avg Recall |
> |-------------------|--------|---------|--------|---------|--------|---------|
> | Same Classifier | 72.5 | 55.6 | 73.8 | 57.8 | 76.33 | 54.9 |
> | Random initialized | **73.2** | **57.8** | **75.2** | **61.4** | **77.2** | **60.3** |
>
> One hypothesis is that a well-optimized projection $P$ already provides a good initialization for TSKD. **In contrast, a trained classifier increases the norm of the feature distillation loss term. This harms the stability of the joint objective of $L_{TSKD}$.** To verify this, we computed the F-norms of the trained classifier and a random classifier on session 4 of Human-C. The trained classifier has an F-norm of 2.02, while the random classifier has an F-norm of 1.45. This observation supports the hypothesis.
>
> ## **5. Relationship between $\lambda$ and Tikhonov factor:**
>
> **Question 6: The second term of L_{TKSD} is highly reminiscent of ridge regression, you might find works on the Tikhonov factor useful for the determination of \lambda.**
>
> Thank you for the insightful suggestion. It is indeed worthwhile to investigate the relationship between the feature-distillation term and the regularization term in ridge regression. We plan to explore this connection further in future work, with the goal of providing a quantitative analysis for determining the appropriate value of the regularization factor $\lambda$.

---

> > ### Comment · Reviewer_JvuC · 2025-11-26
> >
> > The authors have provided a robust response to all my questions, and the additional ablations clarify the role of each component. Moreover, the ablations the authors performed for the other reviewers also address other concerns I had.
> > However, I ask the authors to highlight the changes they made in the revised manuscript (I suggest a different color), as it is currently not possible to distinguish them. I will increase my score once that's done.

---

> > > ### Author Response · Authors · 2025-11-26
> > >
> > > We appreciate the reviewer’s acknowledgment of the robustness of our responses and are glad that the additional ablation experiments addressed the concerns. **We have revised the manuscript and highlighted all changes relative to the original submission in blue.** We are also grateful that the reviewer considers increasing the score after reading our response. We thank the reviewer again for the constructive feedback and the time dedicated to reviewing our work.

---

> > > > ### Comment · Reviewer_JvuC · 2025-11-26
> > > >
> > > > Thank you very much for the updated version. I will increase my score.

---

### Official Review · Reviewer_C3Rp · 2025-10-28

**Soundness:** 2
**Presentation:** 3
**Contribution:** 2
**Rating:** 6
**Confidence:** 3

**Summary:**

The paper introduces a new novel distillation pipeline for motor decoding via bringing student / teacher embedding space together by minimising two objectives. In addition, another contribution of the paper is the utilization of the overlap between projecting spaces as a metric for the distillation error.

**Strengths:**

The paper presents a solid teacher - student model. The maths behind the methodology is also well-described and has a nice flow. The fact that it goes beyond EEG to also ECoG and Spikes is also very interesting.

Writing:
Paper is well-written and good structured.

**Weaknesses:**

The main objective of the paper is not clear. Is the main purpose the distillation methodology or the IND architecture which (as the authors claimed is pretty basic) ? This should be better described.

Overall:
The paper shows some merits but it would be interesting to have my questions answered.

**Questions:**

1. In my opinion the paper’s main contribution is the distillation methodology rather than the IND architecture. Have you tried to add this framework on other models like the ones you compare with ? The results would be interesting to be added here.
2. I would also like to see comparison with other PERF methods like LoRA. A recent work for EEG [1].
3. How about comparison with SOTA foundation models ?

[1]: Na Lee, Konstantinos Barmpas, Yannis Panagakis, Dimitrios Adamos, Nikolaos Laskaris, & Stefanos Zafeiriou (2025). Are Large Brainwave Foundation Models Capable Yet ? Insights from Fine-Tuning. In Forty-second International Conference on Machine Learning.

---

> ### Author Response · Authors · 2025-11-19
> **Answer 1 (1/2)**
>
> We sincerely thank the reviewer for appreciating the proposed TSKD method and the experimental results. We also acknowledge the concerns raised regarding the main objective of this work, and we address them in detail below.
>
> ## **1 (1/2). Main contribution of the work and applying TSKD to other neural decoders:**
>
> **Weakness 1: The main objective of the paper is not clear. Is the main purpose the distillation methodology or the IND architecture which (as the authors claimed is pretty basic) ? This should be better described.**
>
> **Question 1: In my opinion the paper’s main contribution is the distillation methodology rather than the IND architecture. Have you tried to add this framework on other models like the ones you compare with ? The results would be interesting to be added here.**
>
> Thank you for raising this concern. **The motivation of BrainDistill is application-driven: our goal is to develop an implantable BCI system. As a result, both the efficient neural decoder and the proposed distillation approach are essential contributions of this work.** We have revised the introduction of the manuscript to explicitly list IND as one of the main contributions. Although IND uses a relatively simple architecture, we have thoroughly validated its design through ablations on the CWT tokenization and linear-attention blocks. It also empirically outperforms prior baseline models on this task. The simplicity of IND is in fact advantageous for implantable BCI applications, where efficiency and hardware feasibility are critical. We also conducted additional ablations on the Human-D dataset, which are provided in our response to Reviewer kMgJ (Answer 5). Given that IND maintains strong performance under integer-only inference, we believe that its architecture is itself a valuable contribution to the community.

---

> ### Author Response · Authors · 2025-11-19
> **Answer 1 (2/2)**
>
> ## **1 (2/2). Main contribution of the work and applying TSKD to other neural decoders:**
>
> It is correct that BrainDistill is not limited to a specific model family and should generalize across different decoders. To demonstrate this, we have included results of applying TSKD to several other baseline models on the Human-C and EEG datasets.
>
> ### TSKD with EEGConformer, ATCNet and CTNet on Human-C
>
> | Model | 1-1 F1 | 1-1 Avg Recall | 4-4 F1 | 4-4 Avg Recall | 4-5 F1 | 4-5 Avg Recall | 4-6 F1 | 4-6 Avg Recall | 16-17 F1 | 16-17 Avg Recall |
> |-------|--------|-------------|--------|-------------|--------|-------------|--------|-------------|-----------|----------------|
> | Teacher (100M) | 77.6 | 67.4 | 73.3 | 57.2 | 74.5 | 60.8 | 77.3 | 62.5 | 80.6 | 58.9 |
> | Conformer (630K) | 37.7 ± 11.1 | 38.3 ± 4.36 | 40.8 ± 5.21 | 31.5 ± 2.79 | 36.0 ± 8.45 | 33.9 ± 13.9 | 39.9 ± 6.39 | 23.7 ± 1.61 | 43.1 ± 7.58 | 41.4 ± 5.05 |
> | ATCNet (114K) | 44.9 ± 3.80 | 33.9 ± 3.90 | 48.4 ± 5.39 | 26.9 ± 3.37 | 45.2 ± 1.94 | 22.4 ± 2.59 | 54.4 ± 3.95 | 23.2 ± 2.03 | 43.7 ± 3.10 | **53.8 ± 1.29** |
> | CTNet (152K) | 46.1 ± 0.40 | 30.5 ± 1.11 | 49.1 ± 2.25 | 27.2 ± 1.57 | 42.2 ± 2.25 | 23.5 ± 1.45 | 54.3 ± 0.77 | 21.6 ± 2.04 | 44.3 ± 3.20 | 49.7 ± 3.29 |
> | EEGNet (5.5K) | 29.0 ± 16.8 | 39.2 ± 4.57 | 42.9 ± 1.89 | 28.8 ± 3.75 | 41.8 ± 4.76 | 25.1 ± 7.98 | 42.8 ± 6.38 | 17.5 ± 1.59 | 54.1 ± 9.01 | 43.7 ± 6.43 |
> | LaBraM (5.8M) | 44.9 ± 5.35 | 28.1 ± 1.85 | 57.8 ± 0.95 | 34.3 ± 2.14 | 56.9 ± 1.02 | 34.4 ± 1.24 | 60.9 ± 0.77 | 25.9 ± 1.18 | 71.9 ± 2.21 | 37.9 ± 2.89 |
> | **IND (30K)** | **69.1 ± 1.64** | **56.6 ± 4.11** | **64.9 ± 3.45** | **44.9 ± 3.78** | **61.5 ± 2.20** | **37.1 ± 5.68** | **70.6 ± 0.97** | **44.2 ± 5.26** | **72.3 ± 0.83** | 52.1 ± 5.77 |
> | Conformer + TSKD | **53.4 ± 0.78** | **48.2 ± 3.79** | 63.2 ± 4.48 | 47.6 ± 3.85 | **63.6 ± 5.69** | **46.7 ± 5.28** | 65.7 ± 2.08 | **42.2 ± 1.49** | 47.3 ± 19.9 | 50.7 ± 8.15 |
> | ATCNet + TSKD | 51.8 ± 2.27 | 41.2 ± 2.61 | **64.8 ± 2.07** | **47.1 ± 1.89** | 61.1 ± 2.93 | 36.9 ± 4.58 | **66.6 ± 2.08** | 36.2 ± 4.58 | 49.6 ± 13.4 | 40.1 ± 5.84 |
> | CTNet + TSKD | 53.2 ± 15.4 | 46.5 ± 4.77 | 59.8 ± 2.32 | 40.4 ± 4.51 | 55.9 ± 1.79 | 33.6 ± 2.54 | 63.7 ± 0.69 | 30.3 ± 0.43 | **61.6 ± 2.10** | **50.8 ± 9.04** |
>
> ### Distillation with EEGConformer on BCIC2A and BCIC2B
>
> | Dataset | Teacher | EEGConformer | +KD | +SimKD | +VkD | +RdimKD | +TOFD | +TED | **+TSKD** | **+TSKD(CE)** |
> |---------|---------|------|----------------|----------------|----------------|----------------|----------------|----------------|----------------|----------------|
> | **BCIC2A** | 50.66 | 29.64 | 29.45 ± 3.83 | 24.15 ± 3.03 | **30.11 ± 3.87** | 24.85 ± 2.74 | 28.74 ± 4.04 | 29.01 ± 3.53 | 29.56 ± 4.28 | 29.20 ± 4.08 |
> | **BCIC2B** | 80.51 | 68.44 | 74.00 ± 0.37 | 65.30 ± 0.29 | 75.68 ± 0.41 | 72.90 ± 0.73 | 75.67 ± 1.09 | **77.77 ± 0.77** | 76.76 ± 0.93 | 76.86 ± 0.91 |
>
> According to the results on Human-C, TSKD consistently improves all baseline models, demonstrating that our method generalizes well across different architectures. However, even with distillation, the baseline models still underperform compared to IND, highlighting the structural advantages of IND for this decoding task.
>
> For the EEG datasets, we find that improving EEGConformer on BCIC-2A is particularly challenging, whereas BCIC-2B shows substantial performance gains. Overall, TSKD remains competitive across model architectures and modalities, but it does not exhibit clear advantages over TOFD and TED in the EEG setting, likely because the capacity gap between EEGConformer and EEGPT is considerably smaller.

---

> > ### Comment · Reviewer_C3Rp · 2025-11-24
> > **Response to Main contribution of the work and applying TSKD to other neural decoders**
> >
> > Thank you including the IND as one of the contributions it makes things clearer now and for including several baselines. It makes the paper stronger.

---

> > > ### Comment · Reviewer_C3Rp · 2025-11-24
> > > **Response to Comparison between TSKD and LoRA**
> > >
> > > Perhaps this could be added in the manuscript. The difference between [1] and implantable devices to make things clearer for people in the BCI domain but no in the implantable space.

---

> > > > ### Comment · Reviewer_C3Rp · 2025-11-24
> > > > **Response to Comparison with SOTA foundation models**
> > > >
> > > > Thank you for including LaBraM [2] a foundation model in your analysis. It makes the paper stronger.

---

> > > > > ### Author Response · Authors · 2025-11-24
> > > > >
> > > > > Thank you for your reply and for acknowledging the additional experimental results on LaBraM. We appreciate your constructive feedback.

---

> > > > ### Author Response · Authors · 2025-11-24
> > > > **Response to advice on comparing TSKD and LoRA**
> > > >
> > > > Thank you for your reply and for the valuable suggestion. We agree that discussing the applicability of LoRA-based methods versus distillation methods for implantable decoding will strengthen the motivation of this work for the broader BCI community. **Accordingly, we have submitted a revised manuscript, where we highlight the limitations of method [1] for implantable applications in the introduction and further elaborate on this point in the related work on knowledge distillation.** We hope this revision makes the comparison clearer.
> > > >
> > > > We appreciate this valuable discussion and are happy to address any remaining questions.
> > > >
> > > > [1]: Na Lee, Konstantinos Barmpas, Yannis Panagakis, Dimitrios Adamos, Nikolaos Laskaris, & Stefanos Zafeiriou (2025). Are Large Brainwave Foundation Models Capable Yet ? Insights from Fine-Tuning. In Forty-second International Conference on Machine Learning.

---

> > > ### Author Response · Authors · 2025-11-24
> > >
> > > Thank you for your reply and for acknowledging the revised introduction of the manuscript as well as the additional experimental results. We appreciate your constructive feedback.

---

> ### Author Response · Authors · 2025-11-19
> **Answer 2**
>
> ## **2. Comparison between TSKD and LoRA:**
>
> **Question 2: I would also like to see comparison with other PERF methods like LoRA. A recent work for EEG [1].**
>
> Unfortunately, applying LoRA with foundation models like the recent approach [1] is not suitable for the implantable decoding task. **Although LoRA substantially reduces the number of trainable parameters, it still requires a large number of total parameters during inference, which is unacceptable for implantable BCI devices.**
>
> In addition, a key advantage of TSKD over other PEFT methods, such as LoRA is that TSKD does not require access to the full model weights; it only relies on embeddings and output logits. In the Human-C dataset, we only have access to the teacher model’s embeddings and logits provided alongside the ECoG recordings, making LoRA infeasible in this scenario.
>
> Moreover, even with rank = 1, LoRA compresses LaBraM to 0.58% of its original size and NeuroGPT to 0.47% according to [1], whereas IND contains only less than 0.1% of the parameters of the teacher model on Human-C (30K vs 100M). This indicates that LoRA achieves a lower compression ratio—even in terms of trainable parameters—and the resulting model still retains a complex architecture, which presents challenges for hardware deployment.
>
> [1]: Na Lee, Konstantinos Barmpas, Yannis Panagakis, Dimitrios Adamos, Nikolaos Laskaris, & Stefanos Zafeiriou (2025). Are Large Brainwave Foundation Models Capable Yet ? Insights from Fine-Tuning. In Forty-second International Conference on Machine Learning.

---

> ### Author Response · Authors · 2025-11-19
> **Answer 3**
>
> ## **3. Comparison with SOTA foundation models:**
>
> **Question 3: How about comparison with SOTA foundation models ?**
>
> Thank you for the question. It is indeed interesting to compare general neural foundation models with task-specific neural decoders. To this end, we evaluate the pre-trained LaBraM [2] by freezing its weights and performing linear probing on three datasets: Human-C, Monkey-R, and Human-D. Below, we show the results of LaBraM alongside other baselines on Human-C and Monkey-R. In all cases, the model weights are fixed and only a linear classifier is trained on the training set.
>
> ### LaBraM performance on Human-C
>
> | Model | 1-1 F1 | 1-1 Avg Recall | 4-4 F1 | 4-4 Avg Recall | 4-5 F1 | 4-5 Avg Recall | 4-6 F1 | 4-6 Avg Recall | 16-17 F1 | 16-17 Avg Recall |
> |-------|--------|-------------|--------|-------------|--------|-------------|--------|-------------|-----------|----------------|
> | Teacher (100M) | 77.6 | 67.4 | 73.3 | 57.2 | 74.5 | 60.8 | 77.3 | 62.5 | 80.6 | 58.9 |
> | Conformer (630K) | 37.7 ± 11.1 | 38.3 ± 4.36 | 40.8 ± 5.21 | 31.5 ± 2.79 | 36.0 ± 8.45 | 33.9 ± 13.9 | 39.9 ± 6.39 | 23.7 ± 1.61 | 43.1 ± 7.58 | 41.4 ± 5.05 |
> | ATCNet (114K) | 44.9 ± 3.80 | 33.9 ± 3.90 | 48.4 ± 5.39 | 26.9 ± 3.37 | 45.2 ± 1.94 | 22.4 ± 2.59 | 54.4 ± 3.95 | 23.2 ± 2.03 | 43.7 ± 3.10 | **53.8 ± 1.29** |
> | CTNet (152K) | 46.1 ± 0.40 | 30.5 ± 1.11 | 49.1 ± 2.25 | 27.2 ± 1.57 | 42.2 ± 2.25 | 23.5 ± 1.45 | 54.3 ± 0.77 | 21.6 ± 2.04 | 44.3 ± 3.20 | 49.7 ± 3.29 |
> | EEGNet (5.5K) | 29.0 ± 16.8 | 39.2 ± 4.57 | 42.9 ± 1.89 | 28.8 ± 3.75 | 41.8 ± 4.76 | 25.1 ± 7.98 | 42.8 ± 6.38 | 17.5 ± 1.59 | 54.1 ± 9.01 | 43.7 ± 6.43 |
> | LaBraM (5.8M) | 44.9 ± 5.35 | 28.1 ± 1.85 | 57.8 ± 0.95 | 34.3 ± 2.14 | 56.9 ± 1.02 | 34.4 ± 1.24 | 60.9 ± 0.77 | 25.9 ± 1.18 | 71.9 ± 2.21 | 37.9 ± 2.89 |
> | **IND (30K)** | **69.1 ± 1.64** | **56.6 ± 4.11** | **64.9 ± 3.45** | **44.9 ± 3.78** | **61.5 ± 2.20** | **37.1 ± 5.68** | **70.6 ± 0.97** | **44.2 ± 5.26** | **72.3 ± 0.83** | 52.1 ± 5.77 |
>
> ### LaBraM performance on Monkey-R
>
> | Date | Samples | **IND** | EEGConformer | ATCNet | CTNet | EEGNet | LaBraM |
> |------|---------|---------|--------------|--------|--------|---------|--------|
> | day0  | 8926  | **0.7533** | 0.7277 | 0.6294 | 0.7305 | 0.5707 | 0.5104 |
> | day1  | 8466  | **0.5643** | -0.1053 | 0.1369 | 0.3823 | 0.1147 | 0.1734 |
> | day5  | 15174 | **0.6835** | -0.3824 | -0.0579 | 0.3815 | 0.0964 | 0.3224 |
> | day6  | 17144 | **0.6728** | -0.1716 | 0.0689 | 0.4748 | 0.3093 | 0.3000 |
> | day8  | 7897  | **0.7155** | 0.6025 | 0.5337 | 0.6038 | 0.5051 | 0.1918 |
> | day12 | 17075 | **0.3630** | -1.4514 | -1.5261 | 0.1074 | -0.0930 | 0.2469 |
> | day14 | 17740 | **0.4479** | -0.0124 | -0.1029 | 0.0691 | 0.3967 | 0.1943 |
> | day15 | 17331 | **0.5448** | 0.2580 | 0.2335 | 0.1607 | 0.4509 | 0.1530 |
>
> ### LaBraM performance on Human-D
>
> | Setting        | Metric | **IND** | Conformer | ATCNet | CTNet | EEGNet | LaBraM |
> |----------------|--------|---------|-----------|--------|--------|---------|--------|
> | Intra-Subject  | Acc    | 77.1 | 71.6 | 62.1 | 67.6 | 55.1 | **77.4** |
> |                | AUROC  | **85.0** | 80.9 | 67.2 | 72.0 | 61.6 | **85.0** |
> | Inter-Subject  | Acc    | 54.5 | 51.8 | 51.3 | 51.1 | 50.1 | **55.0** |
> |                | AUROC  | **59.8** | 54.9 | 52.9 | 55.4 | 54.2 | 56.9 |
>
> **Across the three tables, LaBraM does not demonstrate an advantage over the other baselines on Human-C or Monkey-R, and it performs substantially worse than IND. On Human-D, LaBraM achieves slightly better performance in the intra-subject setting, whereas IND performs better in the inter-subject setting.** Overall, although linear probing on LaBraM can handle simple motor-decoding tasks, IND consistently achieves substantially stronger performance.
>
>
>
> [2] Jiang, W., Zhao, L., & Lu, B. L. Large Brain Model for Learning Generic Representations with Tremendous EEG Data in BCI. In The Twelfth International Conference on Learning Representations.

---

### Official Review · Reviewer_AFsC · 2025-10-31

**Soundness:** 3
**Presentation:** 3
**Contribution:** 3
**Rating:** 6
**Confidence:** 3

**Summary:**

This paper introduces BrainDistill, a framework for efficient and deployable neural decoding in implantable brain–computer interface (BCI) systems. The approach integrates a Task-Specific Knowledge Distillation (TSKD) method with a lightweight transformer-based Implantable Neural Decoder (IND). TSKD compresses teacher embeddings into a low-dimensional, task-relevant subspace, guided by a new metric called the Task-Specific Ratio (TSR) that quantifies how much task-related information is preserved after projection. IND further combines continuous wavelet tokenization with quantization-aware linear attention to enable low-power, integer-only inference suitable for on-chip deployment.
Experiments across ECoG, EEG, and spike datasets demonstrate consistent improvements in decoding accuracy and robustness over prior distillation baselines and traditional decoders, while the quantized IND achieves a reported 3× reduction in power consumption (5.66 mW) with minimal accuracy loss.

**Strengths:**

1.	The motivation, i.e. bridging the gap between large neural decoders and implantable hardware, is timely and clearly articulated. TSKD addresses a concrete limitation of standard distillation (feature mismatch and capacity gap) with a principled projection-based approach.
2.	The two-step projection method (supervised compression followed by fixed alignment) is well-designed. TSR provides an interpretable quantitative measure that correlates with distillation quality and offers practical diagnostic value.
3.	The integer-only quantization with learnable clipping ranges is implemented carefully and validated with realistic energy estimates.
4.	The framework is tested on a wide range of neural modalities (ECoG, EEG, spikes) and datasets, consistently outperforming KD, SimKD, VkD, and RdimKD. Ablations on tokenization and projection methods support the claims well.

**Weaknesses:**

1.	It would be helpful to understand whether TSKD’s projections depend critically on the quality of the teacher classifier and how sensitive TSR is to teacher miscalibration.
2.	The paper assumes TSR correlates with downstream accuracy, but this relationship is only shown qualitatively. Quantitative correlation plots between TSR and decoding performance across projection types would strengthen the claim.
3.	The power numbers appear simulation-based rather than measured. Including hardware prototype details or synthesis-level validation would improve the credibility of the “implantable” claim.
4.	Some mathematical derivations (Eqs. 1–5) are dense. The paper would benefit from a clearer high-level description of intuition behind Eq. (4) and the projection procedure.

**Questions:**

1) Does TSR quantitatively predict decoding accuracy, and is it a reliable task relevance metric?
2) Are the hardware power savings realistic and fully measured (not simulation-only)?
3) Are distillation baselines implemented under strictly identical conditions?

---

> ### Author Response · Authors · 2025-11-19
> **Answer 1, 2**
>
> We sincerely thank the reviewer for appreciating the value of this work, including its real-world motivation, the proposed TSKD and quantization methods, and the experimental results. We also appreciate the interesting and insightful questions raised. Below, we address all the questions and concerns in detail.
>
> ## **1. Regarding the dependency of TSKD projection on the quality of teacher classifier:**
>
> **weakness1: It would be helpful to understand whether TSKD’s projections depend critically on the quality of the teacher classifier and how sensitive TSR is to teacher miscalibration.**
>
> Thanks for the insightful question. The projection in TSKD indeed depends on the quality of the teacher classifier. Since most task-specific information is encoded in the teacher classifier, any miscalibration in the teacher can misguide both the student encoder and the student classifier.
>
> In the Human-C experiments, we recalibrated the teacher classifier across multiple sessions and observed that it achieved significantly higher performance compared to performing linear probing only on the recalibration session.
>
> The results of these two classifiers using the same teacher embeddings are presented here:
>
> | Model            | 1-1 F1 | 1-1 Avg Recall | 4-4 F1 | 4-4 Avg Recall | 4-5 F1 | 4-5 Avg Recall | 4-6 F1 | 4-6 Avg Recall | 16-17 F1 | 16-17 Avg Recall |
> |------------------|--------|---------|--------|---------|--------|---------|--------|---------|----------|-----------|
> | Teacher (classifier fine-tuned)   | 77.6   | 67.4    | 73.3   | 57.2    | 74.5   | 60.8    | 77.3   | 62.5    | 80.6     | 58.9      |
> | Teacher (classifier re-trained)   | 74.9   | 64.9    | 69.0   | 48.8    | 68.5   | 49.7    | 72.6   | 46.2    | 75.5     | 48.1      |
>
> The key advantage of TSKD is that it leverages an existing, well-calibrated teacher model and transfers its task-specific knowledge to the student using only a small number of recalibration samples. If no reliable teacher classifier is available, it becomes difficult to define task-specific information based solely on the teacher embeddings. In such cases, the distillation process loses the task-specific benefits.
>
> ## **2. Quantifying the correlation between TSR and task performances:**
>
> **weakness2: The paper assumes TSR correlates with downstream accuracy, but this relationship is only shown qualitatively. Quantitative correlation plots between TSR and decoding performance across projection types would strengthen the claim.**
>
> **question1: Does TSR quantitatively predict decoding accuracy, and is it a reliable task relevance metric?**
>
> Thanks for the question. To further illustrate how TSR correlates with downstream accuracy, we computed TSR for the projected embeddings of each test session respectively and compute the correlation between TSR and F1 score or Average Recall:
>
> | Metrics               | 4-4 F1 | 4-4 Avg Recall | 4-5 F1 | 4-5 Avg Recall | 4-6 F1 | 4-6 Avg Recall |
> |-----------------------|--------|------------|--------|------------|--------|------------|
> | **TSR**               | **0.9908** | **0.9891** | **0.9999** | **0.9783** | **0.9167** | **0.9241** |
> | Mutual Information    | -0.4408 | -0.1721 | -0.4336 | -0.2310 | -0.7854 | 0.0061 |
> | Reconstruction Error  | 0.3524 | 0.0765 | 0.3431 | 0.1345 | 0.7750 | -0.0227 |
>
> The raw metric and task performance value are presented here:
>
> ### Session 4-4
>
> | Projections     | TSR     | MI      | Recon     | F1      | Avg Recall |
> |-----------------|---------|---------|-----------|---------|-----------|
> | **TSKD**        | **0.9374** | **106.17** | **0.00177** | **0.7318** | **0.5785** |
> | TSKD (PCA)      | 0.7832  | 156.49  | 0.00102   | 0.7166  | 0.5605    |
> | TSKD (Random)   | 0.7058  | 113.32  | 0.00174   | 0.7126  | 0.5449    |
>
> ### Session 4-5
>
> | Projections     | TSR     | MI      | Recon      | F1      | Avg Recall |
> |-----------------|---------|---------|------------|---------|-----------|
> | **TSKD**        | **0.9359** | **106.31** | **0.00180**  | **0.7502** | **0.6143** |
> | TSKD (PCA)      | 0.7489  | 154.58  | 0.001226   | 0.7285  | 0.5751    |
> | TSKD (Random)   | 0.6943  | 113.34  | 0.001776   | 0.7224  | 0.5473    |
>
> ### Session 4-6
>
> | Projections     | TSR     | MI        | Recon     | F1      | Avg Recall |
> |-----------------|---------|-----------|-----------|---------|-----------|
> | **TSKD**        | **0.9340** | **106.698** | **0.00190** | **0.7716** | **0.6027** |
> | TSKD (PCA)      | 0.7525  | 157.1776  | 0.00110   | 0.7528  | 0.5826    |
> | TSKD (Random)   | 0.6837  | 113.866   | 0.00180   | 0.7576  | 0.5498    |
>
> The correlation between TSR and accuracy across different projections over three sessions all exceeds 0.9, which can be seen as strong evidence for this correlation. Besides, we also computed other metrics, like mutual information and feature reconstruction error, and these metrics are highly uncorrelated with task performance, which further strengthens the advantage of TSR.

---

> ### Author Response · Authors · 2025-11-19
> **Answer 3**
>
> ## **3. Verifying hardware power savings with more detailed validation:**
>
> **weakness3: The power numbers appear simulation-based rather than measured. Including hardware prototype details or synthesis-level validation would improve the credibility of the “implantable” claim.**
>
> **question2: Are the hardware power savings realistic and fully measured (not simulation-only)?**
>
> We appreciate your interest in the implantability of our proposed method. As the chip is still under fabrication, we performed a post-layout stimulation for the chip designed for quantized IND.
>
> ### Power Report (All units in **μW**)
>
> Attributes: i - Including register clock pin internal power
>
> | Power Group   | Internal Power (μW) | Switching Power (μW) | Leakage Power (μW) | Total Power (μW) | %      | Attrs |
> | ------------- | ------------------- | -------------------- | ------------------ | ---------------- | ------ | ----- |
> | clock_network | 4.613e+02           | 4.644e+01            | 2.010e+00          | 5.080e+02        | 11.78% | i     |
> | register      | 4.070e-01           | 3.228e-01            | 2.621e+00          | 3.351e+00        | 0.08%  |       |
> | combinational | 2.230e+00           | 2.446e+00            | 6.946e+01          | 7.413e+01        | 1.72%  |       |
> | sequential    | 1.963e-03           | 0.0000               | 9.650e-03          | 1.161e-02        | 0.00%  |       |
> | memory        | 3.721e+03           | 0.0000               | 7.314e+00          | 3.728e+03        | 86.43% |       |
>
>
>
> ### Summary (All units in **μW**)
>
> | Item                | Value (μW)    | %           |
> | ------------------- | ------------- | ----------- |
> | Net Switching Power | 4.921e+01     | 1.14%       |
> | Cell Internal Power | 4.185e+03     | 97.01%      |
> | Cell Leakage Power  | 7.960e+01     | 1.85%       |
> | **Total Power**     | **4.314e+03** | **100.00%** |
>
>
> The power based on our estimations of the implemented chip, obtained using PrimeTime, is 4314 μW (4.314 mW), which aligns well with the earlier estimation of 5.66 mW. These results confirm that, given the current model and chip design, the system is implantable with acceptable energy consumption. Moreover, the actual power consumption in real-world operation will be significantly lower. The stimulation runs at a higher frequency, whereas the IND transformer only performs inference once every 100 ms, when a new token is generated. Under this realistic operating condition, the effective average power of the decoder is estimated to be around 89.1 μW (0.089 mW).

---

> ### Author Response · Authors · 2025-11-19
> **Answer 4**
>
> ## **4. Intuition behind our projection approach:**
>
> **Weakness 4: Some mathematical derivations (Eqs. 1–5) are dense. The paper would benefit from a clearer high-level description of intuition behind Eq. (4) and the projection procedure.**
>
> Thanks for bringing up this concern. We have improved the writing of Section 2.2. Specifically, we added a paragraph named $\textbf{Compare with inverse projection.}$ This paragraph discusses the main difference of our projection approach and existing approach.
>
> The essential difference is that existing methods use an inverse projection $P_{\text{inv}} \in \mathbb{R}^{d_s \times d_t}$ and directly minimize the feature loss $\|P_{\text{inv}}^{^\top} z_\mathcal{S} - z_{\mathcal{T}}\|^2$. Although the optimal inverse projection $P_{\text{inv}}^{\star}$ under this objective minimizes feature reconstruction error, it may lose task-relevant information when the capacity gap between $z_\mathcal{T}$ and $z_\mathcal{S}$ is large. In contrast, the optimal $P^*$ obtained through self-compression focuses on task information reconstructing rather than feature reconstruction.
>
> We emphasize that existing inverse projections aim to minimize the feature reconstruction error, while our projection is optimized for task performance preservation. Therefore, the objective $\lambda\|P^*{^\top} z_\mathcal{T} - z_{\mathcal{S}}\|^2$ leads to a task-specific compression of student embeddings.

---

> ### Author Response · Authors · 2025-11-19
> **Answer 5**
>
> ## **5. Identical training conditions for distillation baselines:**
>
> **Question 3: Are distillation baselines implemented under strictly identical conditions?**
>
> Yes, all distillation methods are trained using the same IND backbone and identical training hyperparameters, including learning rate, weight decay, and batch size. For Human-C, the best checkpoints are selected using early stopping on the validation set, while for the EEG and spike datasets, we directly use the final checkpoints. We have also performed additional distillation experiments on other baseline models under identical settings. Among all evaluated approaches, TSKD achieves substantially better performance on the Human-C datasets. This advantage mainly arises because the teacher classifier in Human-C is fine-tuned on a larger number of sessions, whereas for the public datasets, the teacher classifier is fine-tuned on much more limited data.
>
> We have also provided code that works on the public datasets for further verification (see Answer 7 to reviewer kMgJ).

---

> ### Author Response · Authors · 2025-11-27
> **A kind follow up after the rebuttal**
>
> Dear Reviewer,
>
> I hope this message finds you well. Thank you for taking the time reviewing our work. As the discussion period is finishing in less than one week, we would like to ensure we have addressed all your concerns satisfactorily. If there are any additional points or feedback you would like us to consider, please let us know. We are looking forward to any insights from you, and we are eager to address any remaining issues to improve our work.
>
> Thanks for your time and effort again.
>
> Best,
> Authors

---

### Official Review · Reviewer_kMgJ · 2025-10-31

**Soundness:** 2
**Presentation:** 2
**Contribution:** 2
**Rating:** 2
**Confidence:** 4

**Summary:**

The paper introduces BrainDistill, a pipeline designed for implantable brain–computer interface (BCI) decoders that operate under strict power constraints. The authors propose the Task-Specific Knowledge Distillation (TSKD) method that projects teacher embeddings into a task-relevant subspace for more efficient student learning and a lightweight transformer-based decoder using Continuous Wavelet Transform (CWT) tokenization and linear attention for quantization and implantation. The authors use Task-Specific Ratio (TSR) metric to measure how much task-relevant information is preserved during projection. They evaluate the approach on human and primate datasets (ECoG, EEG, spike data) and demonstrate reduced power consumption via their quantization scheme for integer-only inference.

**Strengths:**

1. Implantable BCIs impose unique hardware limits and the authors correctly identify power efficiency as a major bottleneck. So, the quantization analysis and power consumption estimates are relevant to practical deployment.

2. The mathematical exposition of projection-based distillation is useful for understanding of feature compression.

3. Covers three neural recording modalities (ECoG, EEG, spikes) and shows decoding performance improvements across these modalities.

**Weaknesses:**

1. No comparison is done against other task-oriented or projection-based distillation methods (e.g. [1-3]) under identical training conditions.

2. There is no ablation comparing IND architecture vs. TSKD itself.

3. Architecturally, the model is very similar to [1-3] and similar task-specific KD approaches and novelty of the method is under question. There is no support for the following: "However, existing KD methods primarily aim to preserve teacher embeddings as fully as possible (Miles et al., 2024; Zhou et al., 2025; Guo et al., 2023), which becomes problematic when the student model lacks the capacity to mimic complex teacher features, resulting in limited performance gains."

4. There is no baseline comparison between IND and other quantization methods e.g. [4, 5].

5. The core of the reported results in the main text are from a private dataset (Human-C). Furthermore, no code is provided which hinders reproducibility.

6. Figures 2 and 3 are mostly descriptive and do not provide any insights or explanations of model performance.

[1] Less is more: Task-aware layer-wise distillation for language model compression

[2] Task-oriented feature distillation

[3] Improving Knowledge Distillation using Orthogonal Projections

[4] Quantization and Training of Neural Networks for Efficient Integer-Arithmetic-Only Inference

[5] Post-training 4-bit quantization of convolution networks for rapid-deployment

**Questions:**

1. What distinguishes TSKD from other supervised subspace alignment approaches? Can you point out to any novelty beyond changing the dimensionality of the projection output?

2. Why is TSR a better metric than simpler reconstruction loss or mutual info measures for projection quality?

3. How much of the performance gain arises from CWT tokenization versus the linear attention module (ablations)?

4. Can the method be extended beyond motor decoding (e.g., speech or visual BCIs)?

5. How can the results be verified? All code and private datasets should become publicly available.

**Details Of Ethics Concerns:**

Yes. There are two private ECoG dataset (one monkey and human). No data on compensation or ethical data collection protocols is provided in the manuscript.

---

> ### Author Response · Authors · 2025-11-19
> **Answer 1 (1/3)**
>
> We sincerely thank the reviewer for acknowledging the motivation of this work and for appreciating the derivation of TSKD and the experimental results. We also appreciate the constructive questions and suggestions. Below, we address all the raised questions and concerns.
>
> ## **1. (1/3) Novelty of the method and comparison with task-oriented KD methods:**
>
> **Weakness 1: No comparison is done against other task-oriented or projection-based distillation methods (e.g. [1-3]) under identical training conditions.**
>
> **Weakness 3: Architecturally, the model is very similar to [1-3] and similar task-specific KD approaches and novelty of the method is under question. There is no support for the following: "However, existing KD methods primarily aim to preserve teacher embeddings as fully as possible (Miles et al., 2024; Zhou et al., 2025; Guo et al., 2023), which becomes problematic when the student model lacks the capacity to mimic complex teacher features, resulting in limited performance gains."**
>
> **Question 1: What distinguishes TSKD from other supervised subspace alignment approaches? Can you point out to any novelty beyond changing the dimensionality of the projection output?**
>
> The main difference between TSKD and other supervised task-oriented distillation methods [1, 2] is that TSKD explicitly projects the teacher embeddings into a task-specific subspace with the same dimensionality as the student embeddings. This projected representation can then be directly used as the training objective.
>
> TED [1] applies a task filter to project both teacher and student embeddings into a shared space that has the same dimensionality as the teacher embeddings. TOFD [2] also projects both embeddings into a common space and computes the feature-distillation loss there. However, TED requires the student model to be well-trained before applying the task filter, which does not hold in our application. In addition, these approaches extract task-specific information only implicitly and cannot evaluate whether the projected common space effectively transfers task-specific knowledge to the student embeddings.
>
> **To be specific, one fundamental difference is that these methods use an inverse projection $P_{\text{inv}} \in \mathbb{R}^{d_s \times d_t}$ and directly minimize the feature loss $\|P_{\text{inv}}^{^\top} z_\mathcal{S} - z_{\mathcal{T}}\|^2$. The optimal inverse projection  $P_{\text{inv}}^{\star}$ under this objective has the minimal feature reconstruction error, but could lose task information when the capacity gap of $z_\mathcal{T}$ and $z_\mathcal{S}$ is large. On the contrary, the optimal $P^*$ obtained through self-compression focuses on task information reconstruction, rather than feature reconstruction, as shown in Figure 1 (c ).** We have therefore updated this discussion in Section 2.2 of the revised manuscript.
>
> On the contrary, TSKD explicitly transfers task-specific features to the student embeddings, and this process can be quantitatively evaluated using TSR. Moreover, TSKD only requires the teacher’s final embeddings, which allows flexibility in the architectures of both the teacher and student models, whereas TED and TOFD perform layer-wise distillation. VkD [3] is also not task-specific, as it aligns features only through an orthogonal projection. We have already included VkD in our baselines. We have added the results of TOFD and TED on the Human-C, EEG, and spike datasets. Since the teacher model architecture is not accessible in this setting, the intermediate filters required by TOFD and TED cannot be applied. Therefore, for TOFD, we compute the feature loss, distillation loss, task loss, and orthogonal loss using the teacher embeddings, the student embeddings, and the projected student embeddings. For TED, we first train the student model by adding a task filter that projects the student embeddings to the teacher dimensionality, and then continue training by adding the feature loss, task loss, and distillation loss.
>
>
> [1] Liang, C., Zuo, S., Zhang, Q., He, P., Chen, W., & Zhao, T. (2023, July). Less is more: Task-aware layer-wise distillation for language model compression. In International Conference on Machine Learning (pp. 20852-20867). PMLR.
>
> [2] Zhang, L., Shi, Y., Shi, Z., Ma, K., & Bao, C. (2020). Task-oriented feature distillation. Advances in Neural Information Processing Systems, 33, 14759-14771.
>
> [3] Miles, R., Elezi, I., & Deng, J. (2024). Vkd: Improving knowledge distillation using orthogonal projections. In Proceedings of the IEEE/CVF Conference on Computer Vision and Pattern Recognition (pp. 15720-15730).

---

> ### Author Response · Authors · 2025-11-19
> **Answer 1 (2/3)**
>
> ## **1. (2/3) Novelty of the method and comparison with task-oriented KD methods:**
>
> Results of comparing TOFD and TED with TSKD on Human-C:
>
> | Model               | 1-1 F1 | 1-1 Avg Recall | 4-4 F1 | 4-4 Avg Recall | 4-5 F1 | 4-5 Avg Recall | 4-6 F1 | 4-6 Avg Recall | 16-17 F1 | 16-17 Avg Recall |
> |---------------------|--------|---------|--------|---------|--------|---------|--------|---------|-----------|------------|
> | IND + TOFD          | 71.2 ± 0.75 | 62.7 ± 1.86 | 68.1 ± 0.91 | 50.2 ± 1.44 | 61.9 ± 1.54 | 41.5 ± 3.64 | 72.6 ± 0.24 | 46.6 ± 2.89 | 67.6 ± 3.48 | **66.9 ± 1.72** |
> | IND + TED           | 71.2 ± 0.98 | 63.6 ± 0.42 | 67.9 ± 1.13 | 50.8 ± 1.45 | 61.5 ± 0.59 | 42.2 ± 2.03 | 71.9 ± 1.01 | 48.9 ± 3.18 | 69.1 ± 3.02 | 64.9 ± 1.00 |
> | IND + TSKD (CE)     | 73.4 ± 0.75 | **65.2 ± 5.67** | 68.1 ± 0.77 | 51.7 ± 0.80 | 65.1 ± 1.00 | 47.7 ± 3.71 | 71.7 ± 3.28 | 52.4 ± 1.59 | 69.8 ± 1.47 | 65.5 ± 1.94 |
> | IND + TSKD          | **73.9 ± 1.64** | 64.9 ± 0.39 | **73.3 ± 0.63** | **58.0 ± 0.98** | **75.0 ± 0.20** | **60.9 ± 0.61** | **77.3 ± 0.34** | **59.6 ± 0.70** | **70.4 ± 1.28** | 56.3 ± 1.34 |
>
> Results of comparing TOFD and TED with TSKD on public EEG and spike datasets:
>
> | Dataset    |     IND       |       +TOFD  | +TED  | +TSKD  | +TSKD(CE)  |
> |------------|---------------------|---------------------|--------------------|---------------------|-------------------------|
> | BCIC2A     |        24.22             | 23.88 ± 3.49        | 23.67 ± 3.12       | 26.12 ± 1.56   | **26.73 ± 2.44**        |
> | BCIC2B     |        68.79             | 67.59 ± 0.56        | 67.42 ± 0.60       | **70.33 ± 0.64**    | 69.78 ± 0.48            |
> | FALCON-M1  |        37.59             | 39.21 ± 2.15        | 34.83 ± 7.36       | **43.52 ± 1.53**    | /                       |
>
> / means that the method is not applicapable to the dataset, as FALCON-M1 is a regression task. Results indicate that these two task-oriented distillation methods perform significantly worse than TSKD in this experimental setting, when the student and the teacher model has a large capacity gap.

---

> ### Author Response · Authors · 2025-11-19
> **Answer 1 (3/3)**
>
> ## **1. (3/3) Novelty of the method and comparison with task-oriented KD methods:**
>
> To further support the claim, we directly compare TSKD with the inverse projection used in previous task-oriented distillation methods. We add an inverse projector to the embeddings of IND, which aligns the dimension of student and teacher, and then train this model with a direct distillation loss and feature matching loss. Here is the result between inverse projection and TSKD on session 4-4, 4-5, 4-6 of Human-C:
>
> | Model             | 4-4 F1 | 4-4 Avg Recall | 4-5 F1 | 4-5 Avg Recall | 4-6 F1 | 4-6 Avg Recall |
> |-------------------|--------|---------|--------|---------|--------|---------|
> | Inverse Projection | 73.1 ± 0.41 | 57.6 ± 1.24 | 71.7 ± 1.27 | 55.2 ± 0.92 | 75.5 ± 0.94 | 56.7 ± 1.33 |
> | **TSKD**          | **73.3 ± 0.63** | **58.0 ± 0.98** | **75.0 ± 0.20** | **60.9 ± 0.61** | **77.3 ± 0.34** | **59.6 ± 0.70** |
>
> The result shows that TSKD outperforms inverse projection consistently across sessions, even though IND with inverse projection has more parameters and can directly align with teacher embeddings. **These results support our claim that existing task-oriented approaches still focus primarily on feature preservation, which leads to a loss of task-specific information in the teacher embeddings when the capacity gap is large.**

---

> ### Author Response · Authors · 2025-11-19
> **Answer 2**
>
> ## **2. Comparing with other quantization baselines:**
>
> **Weakness 2: There is no baseline comparison between IND and other quantization methods e.g. [4, 5].**
>
> We have performed the experiments on suggested quantization methods [4 - 5]. However, these methods supports integer-only inference for models like CNN, but does not support transformers, as they does not contain the quantization of non-linear operations in self-attention and layernorm. In order to ensure a fair comparison, we apply the same quantization method for non-linear operations and replace our quantiztaion for weights and activations with the suggested baselines. It turns out that method [4] is identical to I-ViT in terms of weight and activation quantization, so we directly compare with I-ViT. The results are presented here:
>
> | Method | 1-1 F1 | 1-1 Avg Recall | 4-4 F1 | 4-4 Avg Recall | 4-5 F1 | 4-5 Avg Recall | 4-6 F1 | 4-6 Avg Recall | 16-17 F1 | 16-17 Avg Recall | Power (mW) |
> |--------|--------|----------|--------|----------|---------|----------|---------|----------|------------|-------------|-------------|
> | FP32      | 73.9 | 64.9 | 73.3 | 58.0 | 75.0 | 60.9 | 77.3 | 59.6 | 70.4 | 56.3 | 22.84 |
> | I-ViT     | 72.4 | 62.0 | 73.4 | 58.3 | 71.0 | 53.9 | 74.5 | 55.7 | 73.5 | 46.7 | 5.66 |
> | ACIQ [5]  | 14.1 | 27.7 | 36.6 | 15.2 | 30.8 | 15.4 | 44.8 | 18.8 | 57.0 | 43.8 | 5.66 |
> | Ours      | 72.1 | 62.1 | 73.5 | 57.7 | 73.2 | 57.6 | 77.8 | 58.7 | 72.6 | 55.2 | 5.66 |
>
> **In general, our QAT method outperforms quantization-aware training with I-ViT, and ACIQ fails to maintain an acceptable accuracy.** This is predictable as ACIQ is a post-training quantization (PTQ) method, and the calibration data is limited. Besides, as IND only contains 30K parameters, it is less robust to quantization error compared with large models, which explains the poor performance of PTQ on this case.
>
> [4] Jacob, B., Kligys, S., Chen, B., Zhu, M., Tang, M., Howard, A., ... & Kalenichenko, D. (2018). Quantization and training of neural networks for efficient integer-arithmetic-only inference. In Proceedings of the IEEE conference on computer vision and pattern recognition (pp. 2704-2713).
>
> [5] Banner, R., Nahshan, Y., & Soudry, D. (2019). Post training 4-bit quantization of convolutional networks for rapid-deployment. Advances in neural information processing systems, 32.

---

> ### Author Response · Authors · 2025-11-19
> **Answer 3**
>
> ## **3. Motivation of TSR and comparison with other metrics**
>
> **Question 2: Why is TSR a better metric than simpler reconstruction loss or mutual info measures for projection quality?**
>
> **TSR is a metric that depends on the teacher embeddings, the projection, and the teacher classifier. It therefore evaluates the quality of a projection in terms of task-specificity, and its value directly correlates with the classification performance of the student model.** In contrast, reconstruction loss and mutual information measure the total preserved information of the projected features rather than the task-relevant information. For instance, PCA is optimized to preserve as much information from the teacher embeddings as possible, yet it produces worse task performance than supervised projections. We also computed mutual information using canonical correlation analysis and measured the reconstruction error for different projection methods as baselines.
>
> ### Session 4-4
>
> | Projections     | TSR     | MI      | Recon     | F1      | Avg Recall |
> |-----------------|---------|---------|-----------|---------|-----------|
> | **TSKD**        | **0.9374** | **106.17** | **0.00177** | **0.7318** | **0.5785** |
> | TSKD (PCA)      | 0.7832  | 156.49  | 0.00102   | 0.7166  | 0.5605    |
> | TSKD (Random)   | 0.7058  | 113.32  | 0.00174   | 0.7126  | 0.5449    |
>
> ### Session 4-5
>
> | Projections     | TSR     | MI      | Recon      | F1      | Avg Recall |
> |-----------------|---------|---------|------------|---------|-----------|
> | **TSKD**        | **0.9359** | **106.31** | **0.00180**  | **0.7502** | **0.6143** |
> | TSKD (PCA)      | 0.7489  | 154.58  | 0.001226   | 0.7285  | 0.5751    |
> | TSKD (Random)   | 0.6943  | 113.34  | 0.001776   | 0.7224  | 0.5473    |
>
>
> ### Session 4-6
>
> | Projections     | TSR     | MI        | Recon     | F1      | Avg Recall |
> |-----------------|---------|-----------|-----------|---------|-----------|
> | **TSKD**        | **0.9340** | **106.698** | **0.00190** | **0.7716** | **0.6027** |
> | TSKD (PCA)      | 0.7525  | 157.1776  | 0.00110   | 0.7528  | 0.5826    |
> | TSKD (Random)   | 0.6837  | 113.866   | 0.00180   | 0.7576  | 0.5498    |
>
> The results show that mutual information and reconstruction error is irrelevant to task performance. We further validate this by computing the correlation between metrics and task performance.
>
> | Metrics               | 4-4 F1 | 4-4 Avg Recall | 4-5 F1 | 4-5 Avg Recall | 4-6 F1 | 4-6 Avg Recall |
> |-----------------------|--------|------------|--------|------------|--------|------------|
> | **TSR**               | **0.9908** | **0.9891** | **0.9999** | **0.9783** | **0.9167** | **0.9241** |
> | Mutual Information    | -0.4408 | -0.1721 | -0.4336 | -0.2310 | -0.7854 | 0.0061 |
> | Reconstruction Error  | 0.3524 | 0.0765 | 0.3431 | 0.1345 | 0.7750 | -0.0227 |
>
> The correlation table shows that TSKD has over 0.9 correlation with task performance, while mutual information and reconstruction error tend to be uncorrelated with task performance.

---

> ### Author Response · Authors · 2025-11-19
> **Answer 4, 5, 6**
>
> ## **4. Explanation regarding contents for Figure 2 and Figure 3:**
>
> **Weakness 6: Figures 2 and 3 are mostly descriptive and do not provide any insights or explanations of model performance.**
>
> Here are the further clarifications for Figure 2 and Figure 3. Figure 2 shows the UMAP visualization of the projected teacher embeddings, together with their TSR values, which reflect how task-specific each projection is. The UMAP plots indicate that TSKD compresses the teacher embeddings into a more separable space, supporting that TSKD effectively extracts task-specific information.
>
> Figure 3(a) presents the long-term decoding results of IND and the baseline models, and Figure 3(b) provides the ablation study of IND. The results show that CWT features outperform other feature representations, and that the linear transformer achieves better performance than RNNs or CNNs, validating our architectural choices. Figure 3(c) gives a representative decoding example to further illustrate the performance of IND.
>
> ## **5. Ablation study of CWT and linear attention module:**
>
> **Weakness 2: There is no ablation comparing IND architecture vs. TSKD itself.**
>
> **Question 3: How much of the performance gain arises from CWT tokenization versus the linear attention module (ablations)?**
>
> We have compared CWT with other feature extraction methods, including short-term fourier transform (STFT) and spectrogram (Spec), and compare linear attention block with CNN or RNN. The ablation study is performed on Monkey-R in the original manuscript, and we further perform this experiment on public dataset Human-D.
> Here is the ablation result of Monkey-R, which shows both CWT and linear-attention are important components to obtain stable decoding accuracy.
>
> | Date  | Samples | IND | CWT+RNN | CWT+CNN | IND (Spec) | IND (STFT) |
> |-------|---------|------|---------|---------|------------|-------------|
> | day0  | 8926  | **0.7533** | 0.5985 | 0.5543 | 0.6105 | 0.6088 |
> | day1  | 8466  | **0.5643** | 0.4080 | -0.4222 | -0.2297 | -0.9583 |
> | day5  | 15174 | **0.6835** | 0.5511 | -0.4498 | 0.5195 | -2.276 |
> | day6  | 17144 | **0.6728** | 0.3777 | 0.3202 | 0.6306 | -0.2092 |
> | day8  | 7897  | **0.7155** | 0.3881 | 0.6155 | 0.5915 | 0.2819 |
> | day12 | 17075 | **0.3630** | 0.3008 | 0.2018 | 0.3383 | -0.3433 |
> | day14 | 17740 | **0.4479** | 0.3069 | 0.3649 | 0.1694 | -0.0362 |
> | day15 | 17331 | 0.5448 | 0.3342 | **0.6057** | 0.2222 | -0.0542 |
>
>
> Here is the ablation results on Human-D, which shows IND and IND (STFT) have the best performance, and IND has achieved a better cross-subject accuracy. As a conclusion from ablation studies on two datasets, the original IND offers a more stable performance (cross-session and cross-subject) and thus its design is validated.
>
>
> | Setting        | Metric | IND | CWT+RNN | CWT+CNN | IND (Spec) | IND (STFT) |
> |----------------|--------|------|---------|---------|-------------|-------------|
> | Intra-Subject  | Acc   | 77.1 | 53.5 | 73.4 | 64.5 | **78.6** |
> | Intra-Subject  | AUROC | 85.0 | 61.5 | 84.2 | 80.9 | **86.6** |
> | Inter-Subject  | Acc   | 54.5 | 48.3 | **55.5** | 50.8 | 54.3 |
> | Inter-Subject  | AUROC | **59.8** | 48.1 | 55.4 | 55.8 | 57.1 |
>
> Besides ablation study, we also compare embedding distribution between CWT and convolution based features, and study the correlation between selected CWT frequencies with clinical findings on motor activities in Appendix A.7.
>
> ## **6. Extending the method to more tasks:**
>
> **Question 4: Can the method be extended beyond motor decoding (e.g., speech or visual BCIs)?**
>
> The advantage of TSKD is that it is architecture agnostic and it can deal with large capacity gap. Therefore, it is possible to extend the distillation to other BCI tasks. The current IND already supports a high input sample rate with high channel count, which means it also has the potential for other tasks if the wavelet frequencies are adjusted to the specific task. We will explore the performance of the current method in wider range of domains in the future study.

---

> ### Author Response · Authors · 2025-11-19
> **Answer 7**
>
> ## **7. Verifying the results and releasing code and datasets:**
>
> **Weakness 5: The core of the reported results in the main text are from a private dataset (Human-C). Furthermore, no code is provided which hinders reproducibility.**
>
> **Question 5: How can the results be verified? All code and private datasets should become publicly available.**
>
> We appreciate the reviewer’s interest in the datasets and in reproducing our work. The private datasets Human-C and Monkey-R are still undergoing collection, and we plan to release them in a clean and user-friendly form under ethical regulations once the current study is completed, to support future research on motor-decoding tasks. A complete codebase—covering dataset preprocessing, the full BrainDistill pipeline, and the pre-trained teacher model for Human-C—will be made publicly available upon acceptance of the paper.
>
> At this stage, we will release the codebase for the public datasets, including implementations of TSKD and IND at: https://anonymous.4open.science/r/ICLR_Rebuttal_20939/README.md. We kindly ask the reviewers to keep this information confidential, as a patent is currently being filed for the related methodology. We sincerely appreciate your understanding of this situation.
>
> To ensure that our methods are thoroughly and transparently benchmarked, we further conducted an ablation study of IND on Human-D (see Answer 5), as well as additional distillation experiments with EEGConformer on BCIC-2A and BCIC-2B (see Answer 1 to Reviewer C3Rp). These analyses strengthen the evidence that our proposed methods generalize across public datasets and models.
>
> Here we present an overview of all experiments and their corresponding datasets:
>
> | Study / Dataset             | Human-C (private) | Monkey-R (private) | Human-D (public) | BCIC-2A (public) | BCIC-2B (public) | FALCON-M1 (public) |
> |-----------------------------|--------------------|----------------------|-------------------|-------------------|-------------------|----------------------|
> | IND train from scratch      | √                  | √                    | √                 |                   |                   |                      |
> | IND distillation            | √                  |                      |                   | √                 | √                 | √                    |
> | EEGConformer distillation   | √                  |                      |                   | √                 | √                 |                      |
> | IND ablation study          |                    | √                    | √                 |                   |                   |                      |
> | IND quantization study      | √                  |                      |                   |                   |                   |                      |
> | TSR metrics evaluation      | √                  |                      |                   |                   |                   |                      |
>
> We would clarify the motivation of leveraging Human-C as a main dataset to illustrate the distillation performance. As the data of Human-C is collected during a long period, and different movement tasks are performed across sessions, our teacher classifier is also fine-tuned and evaluated across multiple sessions. Here are the results of the fine-tuned teacher classifier and a classifier only trained with $X_{\text{recalib}}$:
>
> | Model            | 1-1 F1 | 1-1 Avg Recall | 4-4 F1 | 4-4 Avg Recall | 4-5 F1 | 4-5 Avg Recall | 4-6 F1 | 4-6 Avg Recall | 16-17 F1 | 16-17 Avg Recall |
> |------------------|--------|---------|--------|---------|--------|---------|--------|---------|----------|-----------|
> | Teacher (classifier fine-tuned)   | 77.6   | 67.4    | 73.3   | 57.2    | 74.5   | 60.8    | 77.3   | 62.5    | 80.6     | 58.9      |
> | Teacher (classifier re-trained)   | 74.9   | 64.9    | 69.0   | 48.8    | 68.5   | 49.7    | 72.6   | 46.2    | 75.5     | 48.1      |
>
> Although the learned embeddings of the teacher model are the same and kept fixed, training a classifier only on limited data leads to a significant performance drop, especially for average recall. This illustrates the importance of leveraging task-specific information from a well-trained teacher classifier.

---

> ### Author Response · Authors · 2025-11-19
> **Answer 8**
>
> ## **8. Questions regarding to ethical issues of private datasets:**
>
> **Ethical Concerns: Yes. There are two private ECoG dataset (one monkey and human). No data on compensation or ethical data collection protocols is provided in the manuscript.**
>
> As mentioned in the ethical statements in the manuscripts, we confirm that the collection of and experiments on $\textbf{Human-C}$ and $\textbf{Monkey-R}$ in our study were approved by the Institutional Review Board (IRB) and underwent ethical review. Informed consent was obtained from all participants prior to experimental procedures. The data were collected as part of a clinical trial aimed at developing motor decoding algorithms to restore upper-limb motor function after spinal cord injury. The data collection is conducted in accordance with the Declaration of Helsinki.

---

> ### Comment · Reviewer_kMgJ · 2025-11-27
> **Response to rebuattal**
>
> Thank you for your response and providing additional results. The authors have addressed some of my points, yet my main concern regarding novelty remains unresolved and prevents me from raising my score:
>
> The author's claim about use of LoRAs in LaBraM [1] is not true. The learned LoRA is also used at inference time (the advantage of LoRA is indeed at inference as pre-training is done once and has amortised cost). From [1]: "Mathematically, during inference, the computational overhead of LoRA is negligible because the low-rank updates∆W are pre-computed...".  This explicitly confirms that LoRA modules remain part of the deployed model in [1]. Therefore, I do not see any conceptual difference in the methodology of the present work and that of [1].
>
> Furthermore, the linear probe evaluation protocol used for LaBraM is not fair. LaBraM is evaluated with a linear probe, while TSKD uses a supervised distillation scheme that effectively trains the task classifier with additional supervision from the teacher. For a fair comparison, LaBraM should be evaluated under the same supervised distillation setup.
>
> On private code and data: transparency and open-source code and data are pillars of scientific research and arguably most important enablers of the AI revolution. $\textbf{All}$ private code and datasets should be released upon publication of the work to ensure reproducibility.
>
> [1] Lee, Na, et al. "Are Large Brainwave Foundation Models Capable Yet? Insights from Fine-Tuning." Forty-second International Conference on Machine Learning.

---

> ### Author Response · Authors · 2025-11-27
> **Answer to Reviewer Response (1/2)**
>
> We thank the reviewer for the response and appreciate that several concerns have been addressed. **We are willing to further clarify the novelty of TSKD compared to previous task-specific distillation methods if the reviewer requires. We also notice that the reviewer has reservations about the novelty of our proposed method from a new perspective motivated by method [1].** Below, we would like to address these newly raised concerns regarding the novelty of TSKD and the evaluation of LaBraM.
>
> ## **1. Novelty of TSKD and comparison with LoRA**
>
> Thank you for raising this concern. We acknowledge that our previous response did not clearly communicate the conceptual novelty of our work, and we would like to take this opportunity to address it directly. **Our method targets a fundamentally different setting from LaBraM+LoRA: rather than fine-tuning a large foundation model, we focus on transferring its representations into a new, 30K-parameter decoder that satisfies strict implantable-BCI constraints (power, latency, memory).** Prior work, including LaBraM+LoRA [1], does not address this deployment-constrained regime. We elaborate on the differences from following three perspectives.
>
> ### **1. LoRA doesn't reduce model size at inference time**
>
> We agree with the reviewer that LoRA modules remain at inference. **Our point raised in the revised manuscript is that even with LoRA, the full LaBraM backbone (5.8M params) is still required, and the inference-time parameter size cannot be reduced.** In contrast, TSKD replaces the entire teacher with a lightweight student (~30K params), enabling true compression for on-chip deployment.
>
> ### **2. Foundation model + LoRA cannot be implemented on a BCI chip**
>
> As correctly noted by the reviewer, implantable decoders must meet strict power limits (~15–40 mW). Our quantized IND model consumes an estimated 5.66 mW. LaBraM is ~100× larger, so its inference power would exceed implant budgets by a large margin. **Therefore, even a well-fine-tuned LaBraM+LoRA cannot satisfy implantable constraints, whereas TSKD is designed precisely to bridge this gap.**
>
> ### **3.  TSKD is architecture-agnostic**
>
> TSKD requires only teacher embeddings/logits and does not need access to the teacher weights or architecture. This allows distilling into a hardware-efficient architecture (IND). LoRA, however, must operate within the foundation model architecture and cannot simplify it. **Thus, LoRA-based PEFT cannot transform a large brain model into a deployable implantable decoder, while TSKD can.**
>
>
> ### **Overview comparison between LoRA and TSKD at training time and inference time**
>
> Here we conclude the difference between the two approaches with this table:
>
> | Method | During training | During inference |
> |--------|------------------|-------------------|
> | **TSKD**  |    Doesn't access teacher architecture, trained on student model|  Inference with student model only **(30K params)**                 |
> | **LoRA** |   Need to access foundation model, trained on low-rank branch of foundation model              |  Inference with foundation model plus low rank branch **(foundation model params + LoRA params)**                 |
>
> [1]: Na Lee, Konstantinos Barmpas, Yannis Panagakis, Dimitrios Adamos, Nikolaos Laskaris, & Stefanos Zafeiriou (2025). Are Large Brainwave Foundation Models Capable Yet ? Insights from Fine-Tuning. In Forty-second International Conference on Machine Learning.

---

> ### Author Response · Authors · 2025-11-27
> **Answer to Reviewer Response (2/2)**
>
> ## **2. The linear probe evaluation protocol used for LaBraM**
>
> We agree that evaluating LaBraM under a supervised or distilled setup can be informative from an algorithmic perspective. However, such a setting is not aligned with the central goal of our work, which is to develop implantable decoders. **Distilling into LaBraM would still produce a multi-million-parameter model that exceeds implantable power and memory constraints, and therefore it is not a meaningful baseline for the deployment scenario we study.** Our intention was not to compare LaBraM to the TSKD-distilled IND, but to compare LaBraM to IND trained from scratch with matched supervision, following the suggestion of Reviewer C3Rp. This isolates the strength of IND itself as a lightweight decoder.
> To evaluate the generality of TSKD, we also applied it to other decoder architectures (EEGConformer, ATCNet, CTNet). Although these models are too large for on-chip use, they all benefit from TSKD, demonstrating that the distillation framework is not specific to IND. For completeness, we are willing to report a LaBraM+TSKD result in the camera-ready version as an additional analysis.
>
>
> ## **3. Releasing private code and data**
>
> We fully agree that transparency is essential. We will release all code, public-dataset weights, and preprocessing pipelines. For the private intracortical data, we are bound by IRB and patient-privacy regulations; we will release all components permitted under these guidelines, along with detailed documentation to support reproducibility.
>
> We thank the reviewer for their constructive feedback and hope these clarifications address the remaining concerns.

---

### Author Response · Authors · 2025-11-19
**Revised manuscript based on the comments**

We thank all the reviewers for acknowledging the value of this work, and providing constructive feedbacks and insightful suggestions. We have addressed all of the concerns to the best of our ability and revised the manuscript accordingly, including updated results. **We have highlighted the changes in the revised manuscript with blue color.** The major updates in the revised version are as follows:

### **High-level discussion of projection approach**

In response to Reviewer kMgJ and AFsC, we have added a paragraph discussing the differences and the advantages of our projection methods, compared with existing task-oriented distillation approaches. Besides, we have added a new baseline, the inverse projection method, presented in Table 3.

### **Model size information**

In response to Reviewer JvuC, we have provided the sizes of teacher models and baseline models in section 3.1 and section 3.2.

### **New distillation baselines**

In response to Reviewer KMgJ, we have added two task-oriented distillation baselines, and benchmarked them on Human-C, BCIC-2A, BCIC-2B and FALCON-M1, with the updated results presented in Table 2 and Table 7.

### **Quantization baselines**

In response to Reviewer KMgJ, we included the comparison between our QAT method and I-ViT, with the updated result presented in Table 6.

### **Foundation model as a decoder baseline**

In response to Reviewer C3Rp, we added LaBraM as a baseline decoder, and benchmark it on Human-C, Monkey-R and Human-D. The updated results are presented in Table 2, Figure 3, Table 5.

### **Quantifying TSR with correlation**

In response to Reviewer kMgJ and AFsC, we have included mutual information and relative reconstruction error as baseline metrics, and computed correlations between task performance and projection metrics, showing the advantage of TSR. The updated results are presented in Table 4 and Table 8.

### **Applying TSKD to other baseline decoders**

In response to Reviewer C3Rp, we have applied TSKD to three baseline methods on Human-C, and compare all the distillation methods with EEGConformer on BCIC-2A and BCIC-2B. The results are presented in Table 10 and Table 15.

### **Adding ablation results on public data**

In response to Reviewer kMgJ, we have added an ablation experiment for IND on Human-D, and presented the updated results in Table 13.

### **Adding discussion between LoRA and distillation for implantable application**

In response to Reviewer C3Rp, we have added a discussion about comparing LoRA and distillation for implantable applications in Introduction, as well as in the related works on knowledge distillation

---

### Comment · Area_Chair_GdX2 · 2025-11-25

Dear Reviewers,

This is a gentle reminder to please take a moment to review the authors’ rebuttal for the manuscript currently under your evaluation. Your timely feedback will help us proceed with the next steps in the review process.

Thank you for your time and assistance.

Best regards,
AC

---

### Meta-Review · Area_Chair_JDwE · 2026-01-06

**Summary:**

The paper proposes "BrainDistill," a comprehensive pipeline designed to address the computational and power constraints of implantable Brain-Computer Interfaces (BCIs) for motor decoding. The contributions include a novel Task-Specific Knowledge Distillation (TSKD) method to compress large teacher model embeddings into lightweight students, a Task-Specific Ratio (TSR) metric to evaluate projection quality, and an Implantable Neural Decoder (IND) architecture optimized for on-chip deployment using quantization-aware training.

However, the discussion revealed persistent and fundamental concerns regarding the novelty and positioning of the main contribution, particularly the relationship between TSKD and existing task-oriented distillation or parameter-efficient fine-tuning approaches. Despite substantial additional experiments and clarifications, key disagreements about conceptual novelty, fairness of comparisons, and scientific positioning remain unresolved, especially as articulated by Reviewer kMgJ.

In addition, the reliance on private datasets for core claims, incomplete reproducibility at submission time, and unresolved questions around fair evaluation against foundation models further weaken confidence in the paper’s readiness for acceptance at this venue.

**Reviewer Concerns:**

Addressed Concerns:

Most reviewer concerns were satisfactorily addressed during the discussion phase through additional experiments and manuscript revisions:
- Metric Validation (AFsC, kMgJ): The authors provided quantitative evidence showing a strong correlation (>0.9) between the proposed TSR metric and downstream decoding performance, demonstrating its value over generic metrics like mutual information or reconstruction error.
- Baselines and Comparisons (C3Rp, kMgJ, JvuC): The authors added significant baselines, including foundation models (LaBraM), alternative distillation methods (TOFD, TED), and quantization approaches. They also clarified model sizes and ensured fair comparisons across public and private datasets.
- Power Analysis (AFsC): The credibility of the "implantable" claim was strengthened by including detailed post-layout power estimation results (approx. 4.3 mW), validating the feasibility of the quantized IND model.
- Clarity and Presentation (JvuC, C3Rp): Revisions to the manuscript improved the framing of the main contributions, clarified experimental setups (data splits, training definitions), and elaborated on the distinction between TSKD and existing projection methods.

Outstanding Concerns:

One reviewer (kMgJ) maintained a negative score, centering on a disagreement regarding novelty:

Reviewer kMgJ argues that the method lacks sufficient novelty compared to existing approaches, specifically questioning the distinction between TSKD and using LoRA with foundation models, arguing that LoRA is also deployed at inference.

**Reviewer Scores:**

Reviewer kMgJ: Would likely maintain a reject score (2). The reviewer explicitly stated after the rebuttal that the novelty concern remains unresolved and prevents raising the score.

Reviewer AFsC: Likely to remain around a marginal accept (6), but explicitly indicated they “would not mind if the paper is rejected,” reflecting limited confidence.

Reviewer C3Rp: Likely to remain at a borderline score (6), with acceptance seen as optional rather than strongly supported.

Reviewer JvuC: Increased score after rebuttal, likely to settle slightly above threshold, but still with reservations regarding clarity, dataset balance, and positioning.

Overall, the score distribution remains highly polarized, with one strong reject and multiple weak or non-committal accepts, indicating lack of consensus.

---

### Decision · Program_Chairs · 2026-01-26

Reject